# Investigating the ability of satellite occultation instruments to monitor possible geoengineering experiments

Anna Lange [1], Ulrike Niemeier [2], Alexei Rozanov [3], and Christian von Savigny [1]

[1]Institute of Physics, University of Greifswald, Felix-Hausdorff-Str. 6, 17489 Greifswald, Germany
[2]Max Planck Institute for Meteorology, Bundesstr. 53, 20146 Hamburg, Germany
[3]Institute of Environmental Physics, University of Bremen, Otto-Hahn-Allee 1, 27359 Bremen, Germany

**Correspondence:** Anna Lange (anna.lange@uni-greifswald.de)

**Abstract.** Solar radiation management is a method in the field of geoengineering that aims to modify the Earth's shortwave radiation budget. One idea is to inject sulphur dioxide or sulphuric acid into the stratosphere, where sulphate aerosols are then formed. Such experiments can probably be observed, for example, with satellite occultation instruments like SAGE III/ISS. The aim of the current study is to analyse, using MAECHAM-HAM simulations and retrievals with the radiative transfer program SCIATRAN, whether it is possible to detect the formed stratospheric aerosols from emissions of 1 and 2 Tg S/y (sulphur per year) with the currently active satellite occultation instruments, taking into account an error estimate that is as realistic as possible. If these smaller amounts of sulphur are detectable, larger amounts will also be detectable. The calculations show that, considering the natural variability and the assumptions made here, the stratospheric aerosols formed from emissions of 1 and 2 Tg S/y in the quasi steady-state phase can be detected, which is not the case in the first month of the two-year initial phase.

## 1 Introduction

Geoengineering, also known as climate engineering, comprises methods and technologies that aim to deliberately change the climate system in order to mitigate the consequences of climate change (IPCC, 2013). One method is Solar Radiation Management (SRM), which aims at modifying the shortwave radiation budget in the climate system by increasing the planetary albedo (IPCC, 2013). The injection of sulphur dioxide into the stratosphere (stratospheric aerosol injection, SAI) is one idea of SRM (e.g., Budyko, 1977; Crutzen, 2006), mimicking the effects of large volcanic eruptions where extensive amounts of sulphur dioxide reach the stratosphere. The corresponding radiative forcing of the formed sulphate aerosols depends on the injected amount of sulphur dioxide. Another idea is the injection of sulphuric acid ($H_2SO_4$) into the stratosphere (e.g., Vattioni et al., 2019; Janssens et al., 2020; Weisenstein et al., 2022) or carbonyl sulphide (COS) (e.g., Quaglia et al., 2022).

Note that the exact values of the resulting radiative forcing for the same injection rate may vary depending on the model and set up used (e.g., Niemeier and Tilmes, 2017; Laakso et al., 2022). Simulations with MAECHAM-HAM show that an continuous injection of 10 Tg S/y at an altitude of 60 hPa leads to a top of the atmosphere radiative forcing of - (1.79 – 2.06) W/m[2] (Niemeier and Timmreck, 2015). However, it was shown that the forcing efficiency (ratio of sulphate aerosol forcing to injection rate) decreases with increasing injection rate (e.g., Heckendorn et al., 2009; Niemeier and Timmreck, 2015).

SAI is strongly debated, especially the research gaps (e.g., Haywood et al., 2025). However, with progressing climate change and increasing costs and damage, a deployment may become more and more likely. SAI is supposed to be relatively cheap (e.g., Moriyama et al., 2017) and companies or start-ups may see an option to earn money with SAI (e.g, "Make sunsets" company (Make sunsets , 2024)). Therefore, it is very important to be able to observe relatively small amounts of sulphur aerosols in the atmosphere.

One potential way to monitor possible future deployment of SAI is the use of satellite instruments. The SAGE III/ISS (Stratospheric Aerosol and Gas Experiment III) instrument, mounted on the International Space Station (ISS), is a currently operating satellite solar occultation instrument. It measures the attenuation of solar radiation due to scattering and absorption of atmospheric components such as ozone, nitrogen dioxide, water vapour and aerosols. A description of the solar occultation measurement technique can be found in, e.g., McCormick et al. (1979). The instrument observes around 15 sunrises and 15 sunsets in 24 hours and covers a possible latitude range between 70° S and 70° N (NASA, 2022). SAGE III/ISS has 12 spectral channels, of which 9 cover aerosols as target species. The specific 9 wavelengths are: 384, 448, 520, 601, 676, 755, 869, 1021 and 1543 nm (NASA, 2022).

The aim of the current study is to investigate whether it is possible to detect stratospheric aerosols formed from small amounts (in context of possible geoengineering experiments) of sulphur artificially and continuously injected into the stratosphere. Here 1 and 2 Tg S/y, which results in a much lower sulphate injection per time compared to a volcanic eruption with the same injected amount, using a satellite solar occultation instrument, like SAGE III/ISS. It is assumed that if these smaller amounts of sulphur (in context of possible geoengineering experiments) are detectable, larger amounts will also be detectable. Although 1 and 2 Tg S/y do not have a significant climatic effect, about -0.3 and -0.6 W/m$^2$ (Niemeier and Timmreck, 2015), these emission rates were chosen to see whether it is possible to detect even these small amounts, taking into account an aerosol extinction retrieval error estimate that is as realistic as possible. For this purpose, MAECHAM-HAM simulation results for different SRM scenarios were used. These simulation results provided aerosol extinction coefficients at 500 and 550 nm for an altitude range of 10 – 27 km. The aerosol extinction coefficients were used to perform transmission calculations with SCIATRAN from the perspective of a typical solar occultation instrument, which were then used for the subsequent retrieval of the stratospheric aerosol extinction profiles using SCIATRAN. A sensitivity study, an error analysis, and the analysis of natural variability are then carried out to answer the question about the detectability of the geoengineering experiment. Note that although the following analyses examine the detectability for a typical satellite occultation instrument such as SAGE III/ISS, the SAGE retrieval algorithm was not used.

The paper is structured as follows. Section 2 describes the procedure regarding the MAECHAM-HAM simulations, as well as the transmission calculations and the retrievals using SCIATRAN. Section 3 provides an overview of the results, followed by the discussion and conclusions.

## 2 Methodology

### 2.1 MAECHAM-HAM model simulations

The evolution and distribution of sulphate aerosols were simulated with the middle atmosphere (MA) version of the ECHAM general circulation model (GCM) (MAECHAM; (Giorgetta et al., 2006)). MAECHAM was applied with a grid size of about 1.8° x 1.8°, more specific the spectral truncation at wavenumber 63 (T63), and 95 vertical layers up to 0.01 hPa (about 80 km). The model solves prognostic equations for temperature, surface pressure, vorticity, divergence and phases of water. MAECHAM (thereafter ECHAM) was interactively coupled to the prognostic modal aerosol microphysical Hamburg Aerosol Model (HAM) (Stier et al., 2005), which calculates the formation of sulphate aerosols including nucleation, accumulation, condensation and coagulation, and their removal processes by sedimentation and deposition. HAM has been adapted to stratospheric conditions, considering, e.g., that the temperature range for nucleation processes is valid for low stratospheric temperature, and that the width of the mode bins (sigma) differs from the sigma in the troposphere (Kokkola et al., 2009; Niemeier et al., 2009; Niemeier and Timmreck, 2015). We prescribe reactive gases (e.g., ozone, nitrogen oxides, hydroxyl radical–OH) and photolysis rates of carbonyl sulphide (OCS), $H_2SO_4$, $SO_2$, $SO_3$ and $O_3$ on a monthly average basis. The initial conversion of $SO_2$ to $H_2SO_4$ is simulated with a simple stratospheric sulphur chemistry scheme, which is applied above the tropopause (Timmreck, 2001; Hommel et al., 2011).

The model simulates the related dynamical processes and generates the quasi biennial oscillation in the stratosphere. The model is not coupled to an ocean model. We run the model with prescribed sea surface temperatures (SST) and sea ice, set to climatological values (Hurrell et al., 2008), averaged over the AMIP (Atmospheric Model Intercomparison Project) period 1950 to 2000 and does not change due to SAI. Besides the prescribed SSTs the model is running freely. It calculates the dynamical processes following the equations in the GCM. No nudging, relaxing the prognostic variables towards an atmospheric reference state to, e.g., ERA5 data, is applied.

The direct radiative effect of sulphate aerosol is included for both solar (shortwave, SW) and terrestrial (longwave, LW) radiation and coupled to the ECHAM radiation scheme. The model diagnoses the instantaneous aerosol radiative forcing via a double radiation call, once with aerosols and once without aerosols. The aerosols absorb near-infrared and LW radiation, which heats the stratosphere and dynamically influences the resulting processes. This model has already been successfully applied in previous climate engineering studies (Niemeier and Schmidt, 2017; Niemeier et al., 2020). The cooling of the troposphere is included as well. As we use climatological SSTs the surface cooling of the sulfate aerosols is seen in the model over land only, not over the oceans. The latter would requires a coupled ocean model.

Simulations were performed with 1 and 2 Tg S/y. $SO_2$ was injected continuously at an altitude of 60 hPa ($\approx$ 19 km) into one grid box 2.8° x 2.8° centred at the equator at 121° E. This study uses data of the initial phase, i.e. the first two years, and of the quasi steady-state phase, an average over three years (years 12 to 15). We performed a single simulation over several years. The injections for SAI ran for 15 years. For our study, we took three years at the end of these simulations and averaged them over time. This is similar to previous simulations and publications, e.g. Niemeier et al. (2020) and Weisenstein et al. (2022), where three-year averages were also used. Fig. A1 illustrates the time series of the global sulphate burden showing that the

steady-state phase is reached after two years. We used the early phase to include sulphate level below the steady-state level to see if we could detect sulphate even earlier. At this point, the goal was not to use a stabilized result. The aim was to find a lower threshold at which detection would be possible.

The model output of the 95 levels was interpolated to provide aerosol extinction coefficients at 500 and 550 nm between 10 – 27 km, in 1 km steps. The provided aerosol extinction coefficient profiles are monthly averages.

In the following, the months of January and July are analysed for the latitude range of 85° S to 85° N, in 10° steps.

## 2.2 Simulations and Retrievals with SCIATRAN

The SCIATRAN radiative transfer model was developed by the Institute of Environmental Physics at the University of Bremen, Germany (Rozanov et al., 2014). SCIATRAN was originally designed for satellite-based data retrieval. More information can be found at https://www.iup.uni-bremen.de/sciatran/ (last access: 30 April 2025).

Based on the ECHAM simulation results, in this case aerosol extinction coefficients for an altitude range of 10 to 27 km for 500 and 550 nm, the aerosol extinction coefficients at 520 nm were first calculated using the Ångström parameterisation. Using the aerosol extinction coefficients at 520 nm, the corresponding transmission values for a solar occultation observation geometry were simulated with the SCIATRAN radiative transfer model, which were then used for the retrieval with SCIATRAN to retrieve the corresponding aerosol extinction profiles. More details are provided below.

### 2.2.1 Transmission calculations

Using the aerosol extinction coefficients at 500 and 550 nm from the ECHAM model simulations, the aerosol extinction coefficients at 520 nm were first calculated using the Ångström parameterisation, since this is one of the spectral channels of SAGE III/ISS for aerosols as target species (NASA, 2022). For this purpose, the Ångström exponents $\alpha$ were calculated from the given aerosol extinction coefficients $k_{500}$, $k_{550}$ and the aerosol extinction coefficients at 520 nm ($k_{520}$) were then determined as follows (Ångström, 1929):

$$k_{520} = k_{500} \left( \frac{520}{500} \right)^{-\alpha} \tag{1}$$

with:

$$\alpha = -\frac{\ln \left( \frac{k_{550}}{k_{500}} \right)}{\ln \left( \frac{550}{500} \right)} \tag{2}$$

The exact values of $\alpha$ depend, e.g., on the considered month, latitude and injection scenario. For example, for January, 45° N and background (0 Tg S/y), the values are between $\approx$ 1.0 and 2.2, depending on the altitude. To calculate the corresponding transmission values through the Earth's atmosphere at 520 nm, the SCIATRAN radiative transfer model (version 4.7) was used (Rozanov et al., 2014). In the transmission modelling mode (solar/lunar occultation mode) the direct solar radiation transmitted through the spherical Earth's atmosphere is simulated. Atmospheric refraction was also considered in the simulations. The vertical profiles of pressure, temperature and trace gases required for the simulations were taken from the implemented

climatological database, which is based on a 3-D chemical transport model (Sinnhuber et al., 2003). Further input parameters for the transmission calculations with SCIATRAN are listed in Tab. 1 below.

**Table 1.** Input parameters for the transmission calculations with SCIATRAN.

| Parameter | Setting |
|---|---|
| Height grid | 0 - 100 km, 1 km steps |
| Tangent height grid | 10 - 60 km, 2 km steps |
| Vertical field of view | 0.0083 deg |
| Trace gases | $H_2O$, $O_2$, $N_2O$, $NO_3$, $NO_2$, $CO_2$, $O_3$, $SO_2$ |
| Total ozone column (TOC) | 300 DU (Dobson units) |

120

The input parameters, such as those relating to the viewing geometry, are selected so that they correspond to an imaginary satellite occultation instrument, like SAGE III/ISS, with a satellite altitude of 400 km. Output of the simulations with SCIATRAN are transmission values at 520 nm for the tangent heights of $10 - 60$ km, in 2 km steps (compare Tab. 1).

### 2.2.2 Retrieval

For the retrieval of the extinction profiles at 520 nm, the retrieval algorithm in SCIATRAN 4.7 was used, which is not publicly available. A description of the retrieval algorithm can be found in Rozanov et al. (2011). The retrieval approach is the regularised inversion with the optimal estimation method.

The linearised inverse problem is formulated as follows:

$$y = F(x_a) + K(x - x_a) \qquad (3)$$

with $y$ as the data vector (or measurement vector), containing the logarithms of the transmission values at 520 nm, $F$ is the radiative transfer operator, $x_a$ the a priori state vector, $K$ the weighting function matrix and $x$ the state vector (to be retrieved). The a priori state vector $x_a$ is kept constant over the iteration steps. The (approximate) solution of the inverse problem (Eq. 3) is obtained by minimising the following equation:

$$\|F(x_a) + K(x - x_a) - y\|^2_{S_\epsilon^{-1}} + \|(x - x_a)\|^2_{S_a^{-1}} \qquad (4)$$

here $S_\epsilon$ is the noise covariance matrix and $S_a$ the a priori covariance matrix. Since the inverse problem is not linear, the Gauss-Newton iterative approach is used to formulate the solution for each iteration step $x_{i+1}$ as follows:

$$x_{i+1} = x_a + \left(K_i^T S_\epsilon^{-1} K_i + S_a^{-1}\right)^{-1} K_i^T S_\epsilon^{-1} \left(y - F(x_i) + K_i(x_i - x_a)\right) \qquad (5)$$

More detailed information on the retrieval algorithm can be found in Rozanov et al. (2011), Sect. 3.4.2.

Table 2 shows the relevant input parameters for the retrieval. Note that for the purpose of clarity, some of the input parameters that are already listed in Tab. 1 have also been added here. In addition, the calculated transmission values at 520 nm (Sect.

2.2) for the corresponding latitudes and months were used as data input. The retrieval was restricted to the altitude range of the aerosol extinction coefficients provided (see Sect. 2.1), i.e. from 10 to 27 km. Background aerosol extinction coefficient profiles at 520 nm corresponding to the latitude and month were used as a priori information. These profiles originate from the ECHAM model simulations for 0 Tg S/y (compare Sect. 2.1).

**Table 2.** Relevant input parameters for the retrievals with SCIATRAN.

| Parameter | Setting |
|---|---|
| Height grid | 0 - 100 km, 1 km steps |
| Tangent height grid | 10 - 60 km, 2 km steps |
| Vertical field of view | 0.0083 deg |
| Total ozone column (TOC) | 300 DU |
| Apriori variance | 30 % |
| Convergence criterion | 2 % |
| Signal to noise ratio (SNR) | 1000 |

As already mentioned in Sect. 2.1, the retrievals were carried out for the latitudes 85° S to 85° N, in 10° steps for January and July (for the quasi steady-state phase data) and January (for the initial phase data). Output of the retrieval with SCIATRAN here is, among other data output files, the retrieved aerosol extinction profile at 520 nm.

## 3   Results and discussion

Figure 1 shows the extinction coefficients at 550 nm for January (Jan) to December (Dec) based on the ECHAM model simula-
tions for the injection of 1 Tg S/y for the quasi steady-state phase and the initial phase, demonstrating the differences between these two phases. The upper panel (a) of Fig. 1 displays ECHAM simulation results of the first year of the initial phase, and the lower panel (b) of the quasi steady-state phase. As previously described, the initial phase data includes the first two years and the quasi steady-state phase data an average over three years (years 12 to 15).

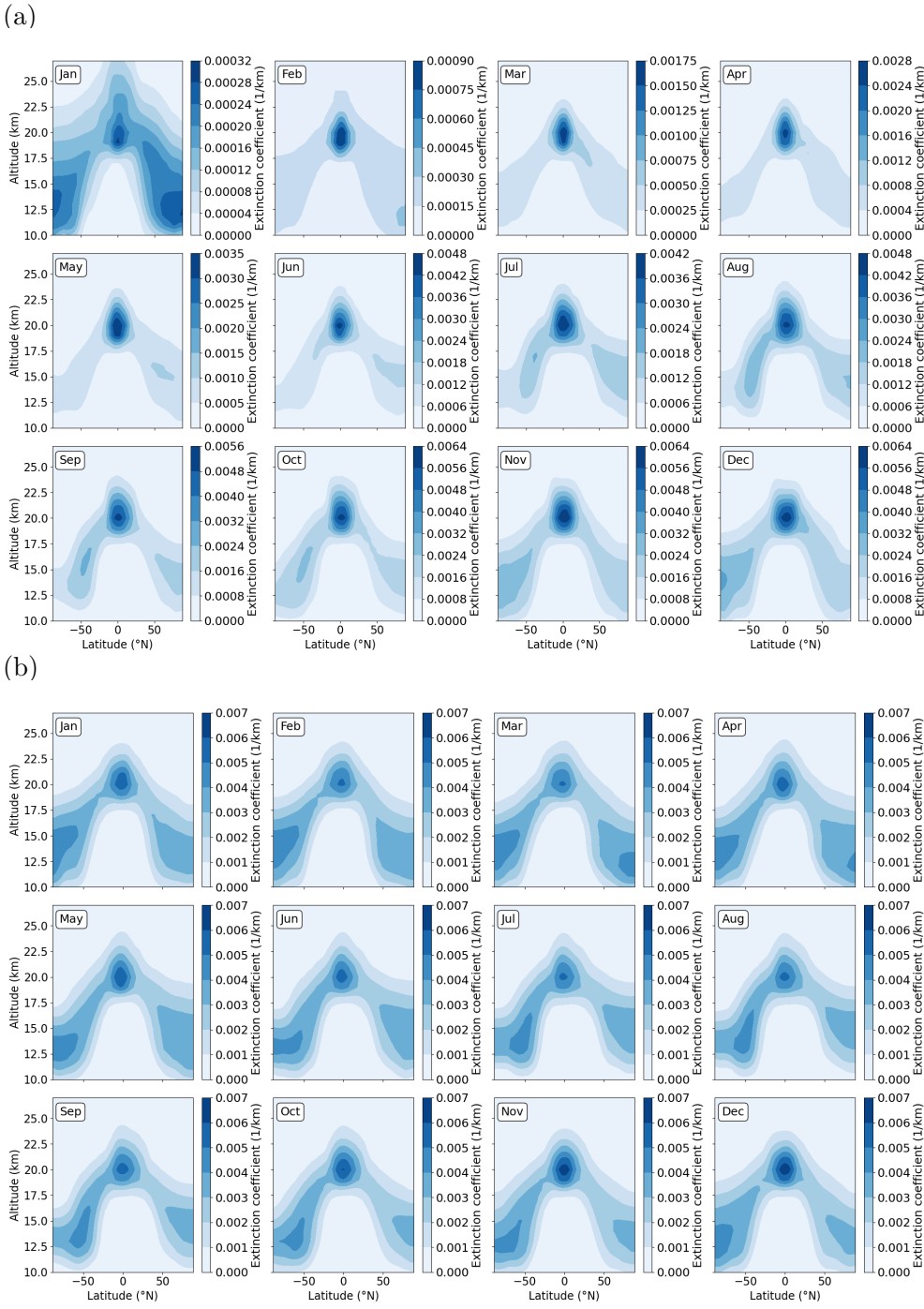

**Figure 1.** Extinction coefficients at 550 nm for January (Jan) to December (Dec) based on the ECHAM model simulations for the injection of 1 Tg S/y. Upper panel (a): Results for the initial phase (first year shown). Lower panel (b): Results for a quasi steady-state phase. Note that the values of the colourbars vary between the upper panel (a) and lower panel (b).

## 3.1 Sensitivity study

In order to answer the question of whether it is possible to detect the sulphate aerosols formed from relatively small sulphur injections of 1 and 2 Tg S/y with the currently active satellite occultation instruments, an error analysis was carried out. The aim is first to estimate the individual errors in aerosol extinction caused by uncertainties or incorrect knowledge of relevant input parameters, e.g. ozone, pressure, temperature (compare Tab. 3) so that an overall error estimate can then be made. For this purpose, the total ozone column was increased by 2 % , the temperature by 2 K and the corresponding pressure was adjusted so that the air density remains constant. Besides this, the pressure was increased by 2 % and the pointing error was taken into account by shifting the tangent height grid upward by 100 m. The corresponding references are listed in Tab. 3. In addition, the noise error taken from the noise covariance matrix was included in the error analysis.

**Table 3.** Reference and modified settings for the sensitivity study.

| Parameter | Reference setting | Modified setting |
|---|---|---|
| Total ozone column (TOC) | 300 DU | + 2 % (e.g., Garane et al., 2019) |
| Temperature and pressure (constant air density) | - | + 2 K (e.g., Nowlan et al., 2007; Langland et al., 2008) |
| Pressure | - | + 2 % (e.g., Nowlan et al., 2007; Langland et al., 2008) |
| Pointing error | - | + 100 m tangent height grid (e.g., Bramstedt et al., 2012) |

Figure 2 shows the retrieved aerosol extinction profiles at 520 nm with reference settings (black line) and modified settings (red line) exemplarily for 1 Tg S/y, 5° N, January based on the ECHAM model simulation results of the quasi steady state phase (left column) and the resulting relative difference (individual errors) (right column). The following parameters are exemplarily shown: (a) Total ozone column, (b) Pressure and (c) Pointing error. More Figs. for different latitudes can be found in the appendix.

The relative differences were determined as follows:

$$r = \frac{x - \mathrm{ref}}{\mathrm{ref}} \cdot 100\% \tag{6}$$

where ref is the retrieved aerosol extinction profile based on the reference settings (black lines in the left column of Fig. 2) and $x$ the retrieved aerosol extinction profile based on the modified settings (red lines in the left column of Fig. 2).

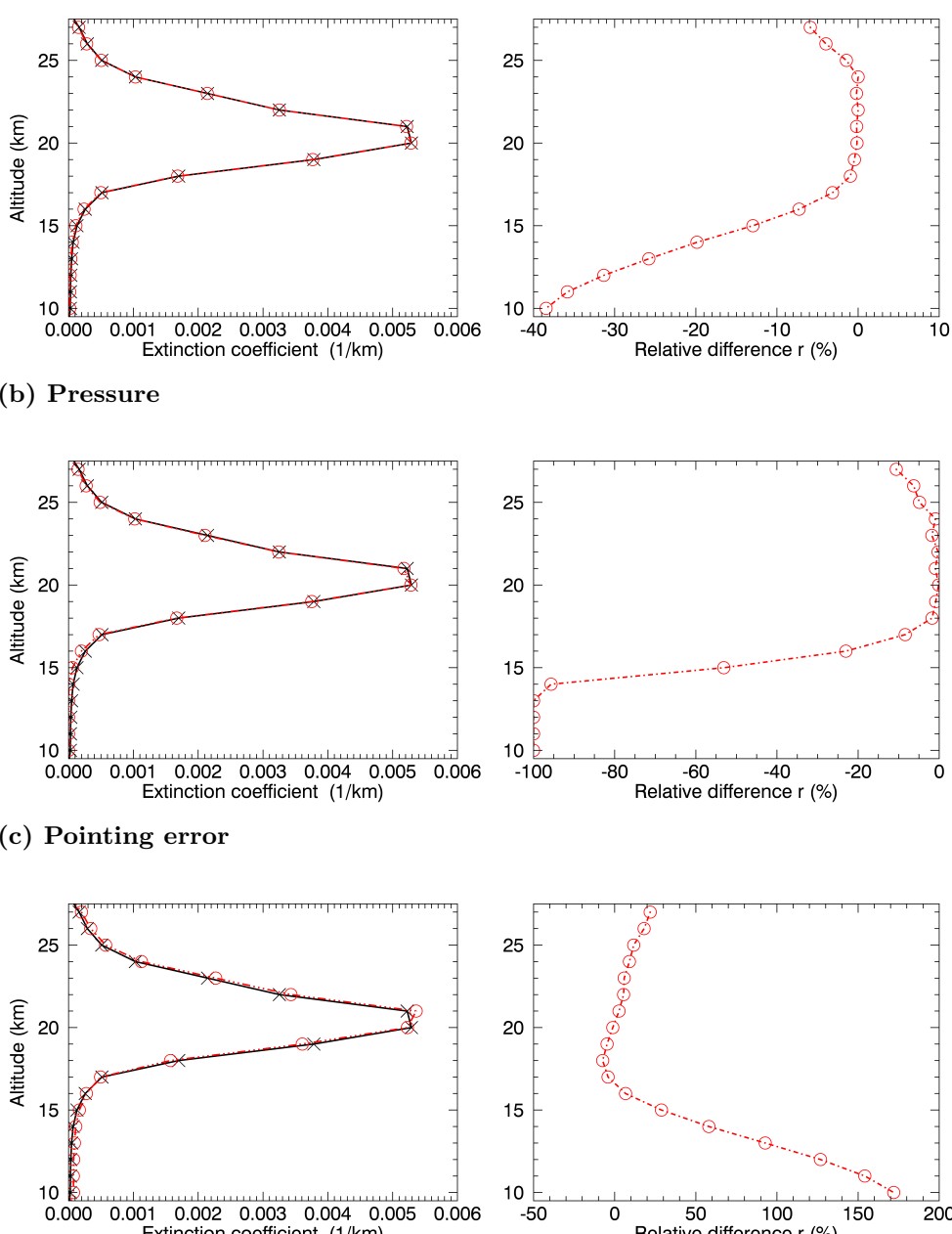

**Figure 2.** Left column: Retrieved aerosol extinction profiles at 520 nm with reference settings (black line) and modified settings (red line) for 1 Tg S/y, 5° N latitude, January based on the ECHAM model simulation results of the quasi steady state phase. Right column: Corresponding relative difference $r$. Both for perturbed (a) Total ozone column, (b) Pressure and (c) shifted tangent height grid.

It should be noted at this point that for the retrievals based on the initial phase data, the errors for the background case are used in the following, based on the assumption that these errors (relative differences) are larger, which is why an error analysis was also carried out for the background case. In order to determine the total error for each height, the subsequent approach was

175 used.

$$\sigma^2_{total} = \sigma^2_{Total\ ozone\ column} + \sigma^2_{Temperature} + \sigma^2_{Pressure} + \sigma^2_{Pointing\ error} + \sigma^2_{Noise} \qquad (7)$$

The individual errors (relative differences) (compare Eq. 6) of a certain height were added up quadratically. This approach is based on the assumption of random and statistically independent error sources and a linear dependence of the derived aerosol extinction profiles on the parameters. The term "total error" used in the following refers to the square root of Eq. 7.

## 3.2 Quasi steady-state phase data

Figure 3 shows the graphical illustration of the total errors in % (Eq. 7) for 1 Tg S/y (upper panels) and 2 Tg S/y (lower panels), January (left panels) and July (right panels). As described in Sec. 2.1 the quasi steady-state phase data is an average over three years (here years 12 to 15). he total errors at the altitude of the SAI injection (here $60\,\text{hPa} \approx 19\,\text{km}$) depend on the emission rate, month and latitude. For 1 Tg S/y, January it is 10 % (highest value) ($85°\,\text{N}$) and 3 % (lowest value) ($25°\,\text{N}$, $25°\,\text{S}$, $35°\,\text{S}$), in July it is 47 % (highest value) ($85°\,\text{S}$) and 3 % (lowest value) ($35°\,\text{S}$). For 2 Tg S/y, January it is 8 % (highest value) ($85°\,\text{N}$) and 3 % (lowest value) ($25°\,\text{N}$, $25°\,\text{S}$, $35°\,\text{S}$), in July it is 47 % (highest value) ($85°\,\text{S}$) and 3 % (lowest value) ($35°\text{N}$, $25°\,\text{S}$). Overall, the total errors for 1 Tg S/y are greater than for 2 Tg S/y, which is consistent with the expectations, as the signal is stronger at a higher injection rate such as 2 Tg S/y and the total errors are therefore smaller.

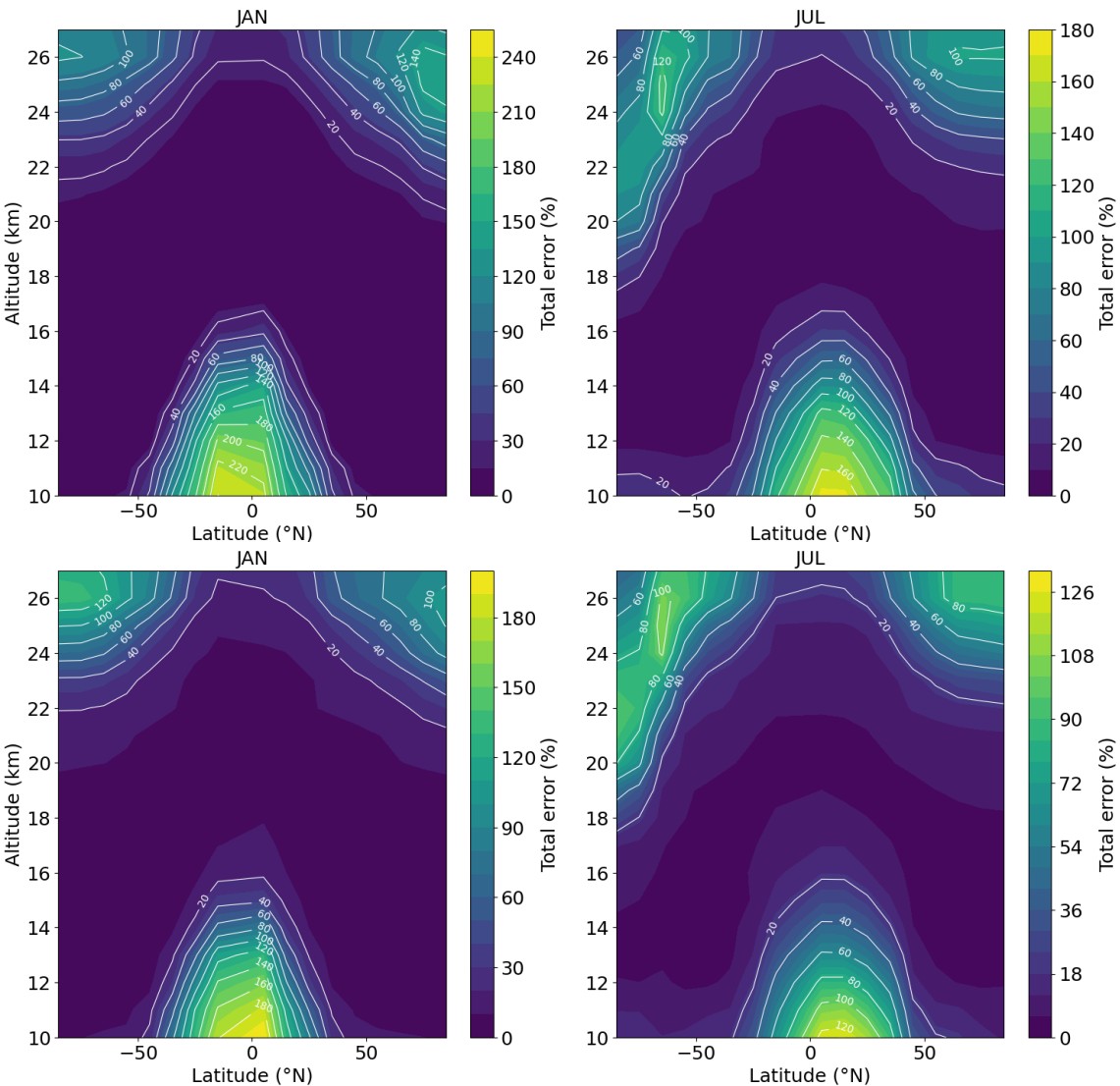

**Figure 3.** Total errors (%) for 1 Tg S/y (upper panels) and 2 Tg S/y (lower panels), January (left panels) and July (right panels) of the quasi steady-state phase.

Figures 4 and 5 show the retrieved aerosol extinction profiles at 520 nm (black dashed lines) for 1 Tg S/y (Fig. 4) and 2 Tg
S/y (Fig. 5), both for January (left column) and July (right column) including the corresponding total errors (red dashed lines), the background profiles (black solid lines) and true profiles (purple solid lines) both at 520 nm (ECHAM model simulation results) for 65° N (upper panels), 15° N (middle panels) and 65 ° S (lower panels).

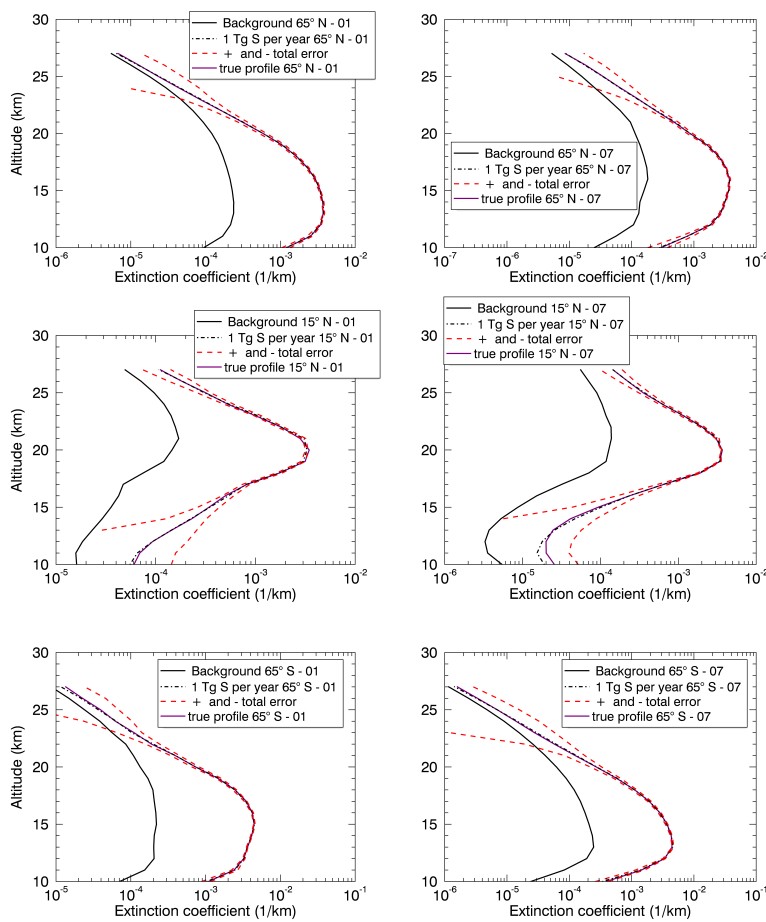

**Figure 4.** Retrieved aerosol extinction profiles at 520 nm for 1 Tg S/y, January (left column) and July (right column) of the quasi steady-state phase including total errors, background profiles (520 nm) and true profiles (520 nm) (ECHAM simulation results). Latitudes: 65 °N (upper panels), 15 °N (middle panels), and 65 °S (lower panels).

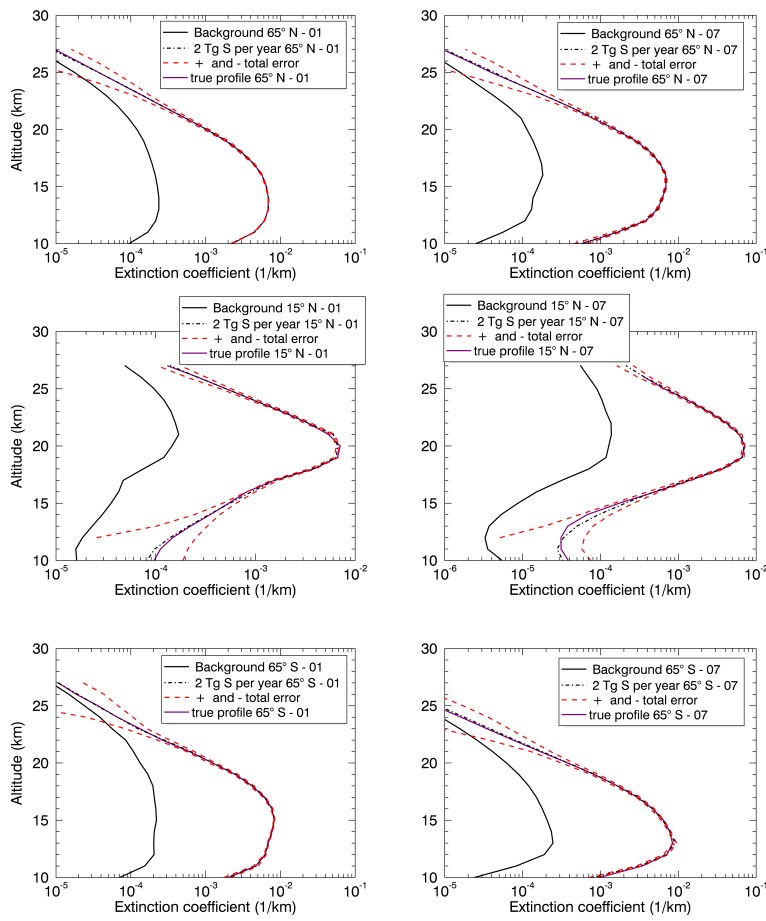

**Figure 5.** Retrieved aerosol extinction profiles at 520 nm for 2 Tg S/y, January (left column) and July (right column) for the quasi steady-state phase including total errors, background profiles (520 nm) and true profiles (520 nm) (ECHAM simulation results). Latitudes: 65 ° N (upper panels), 15 ° N (middle panels), and 65 ° S (lower panels).

The basic idea is that the artificial aerosol enhancement is observable in a certain altitude range, if the background profile is outside the error range of the extinction profiles of 1 and 2 Tg S/y (red dashed lines in Figs. 4 and 5). It means that the change due to the additional emissions of 1 or 2 Tg S/y is large enough to be distinguishable from the background. For 1 Tg S/y (Fig. 4), this is the case for $65\,°$ N in January and July (upper panels) from $\approx 10 - 22$ km; for $15\,°$ N, in January, July (middle panels) from $\approx 14 - 27$ km; and for $65\,°$ S in January and July (lower panels) from $\approx 10 - 22$ km. Note that only a part of the results at different latitudes are shown here, $65\,°$ N, $15\,°$ N, and $65\,°$ S. For 2 Tg S/y (Fig. 5), the artificial aerosol enhancement is observable for $65\,°$ N in January and July (upper panels) from $10 - 24$ km; for $15\,°$ N, in January, July (middle panels) from $\approx 13 - 27$ km; and for $65\,°$ S in January from $\approx 10 - 23$ km and in July from $\approx 10 - 21$ km (lower panels). In both cases, i.e. 1 and 2 Tg S/y, it is possible to observe these emissions for the latitudes and months shown here. In the following, the SAOD at 520 nm and the corresponding total error were determined. To calculate the SAOD, the latitude-dependent tropopause heights from SAGE II data (2002 – 2004) for January (2002) and July (2004 - due to the lack of latitude coverage in 2002 and 2003) were used (NASA, 2012), which is why only latitudes from - 55 to $55\,°$ N can be shown here. The upper limit remains at 27 km corresponding to the altitude range of the given aerosol extinction coefficients (see Sect. 2.1). The total error for the SAOD was calculated using Gaussian error propagation. Figure 6 shows results for latitudes from - 55 to $55\,°$ N for January (left column) and July (right column), with 1 Tg S/y (upper panels) and 2 Tg S/y (lower panels), including the corresponding total errors, background cases and true profiles (ECHAM simulation results).

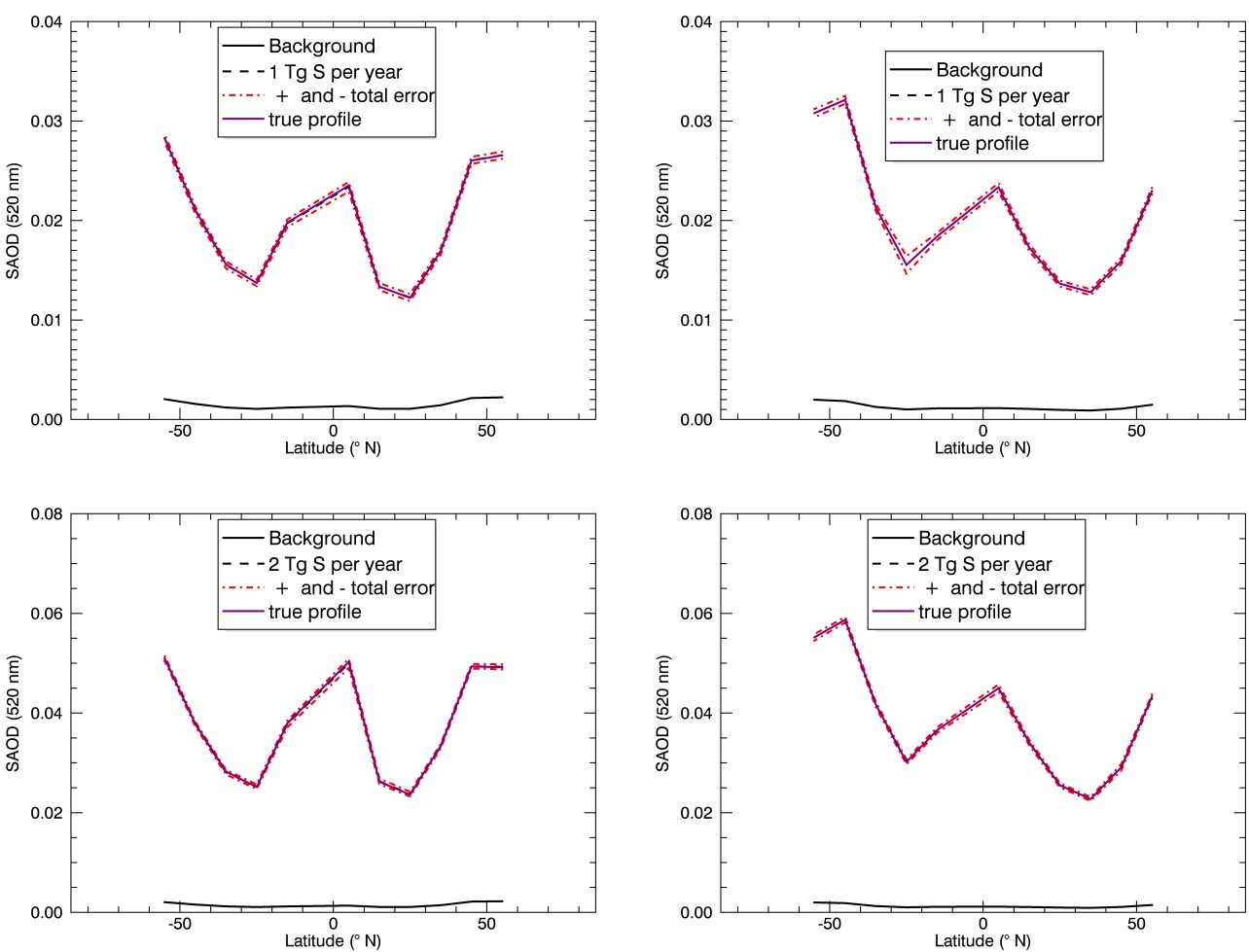

**Figure 6.** SAOD (520 nm) over latitude (° N) for 1 Tg S/y (upper panels) and 2 Tg S/y (lower panels), and for January (left column) and July (right column), including the corresponding total errors, background cases and true profiles (ECHAM simulation results).

All subplots show a certain symmetry in the SAOD about the equator, as well as maxima of the SAOD at mid and high latitudes and minima in the subtropics. The maximum in SAOD near the equator is due to the fact that 1 or 2 Tg S/y are continuously injected in this region (compare Sect. 2.1). The minima of the SAOD in the subtropics and maxima at mid and high latitudes are due to the latitude dependence of the tropopause height, which is higher in the tropics (about 18 km) than at the mid and high latitudes (about 8 km). Since in all cases, i.e. 1 and 2 Tg S/y for January and July, the background 'profile' is outside the error range, it can be concluded that the emissions of 1 and 2 Tg S/y can be observed for the latitudes and months considered here. This means that, under the assumptions made here, it is probably possible to detect the formed stratospheric

aerosols with a satellite occultation instrument for the conditions described here. However, it should be noted that the natural variability has not yet been taken into account at this point and will be discussed below.

### 3.3 Initial phase data

As mentioned above, the total errors from the background case are used for the error analysis of the retrieval of the first month of the initial phase. Figure 7 shows the total errors in % for the background case, i.e. 0 Tg S/y, January. The aim is to see whether it is possible to observe the emission of 1 Tg S/y in the first month of the initial phase subplot (Jan) of panel (a) in Fig. 1). Comparing the total errors with those for 1 and 2 Tg S/y in the quasi steady-state phase (see Sect. 3.2), the total errors in the background case are clearly larger. Also in this case, the total errors vary with altitude and latitude (see Fig. 7).

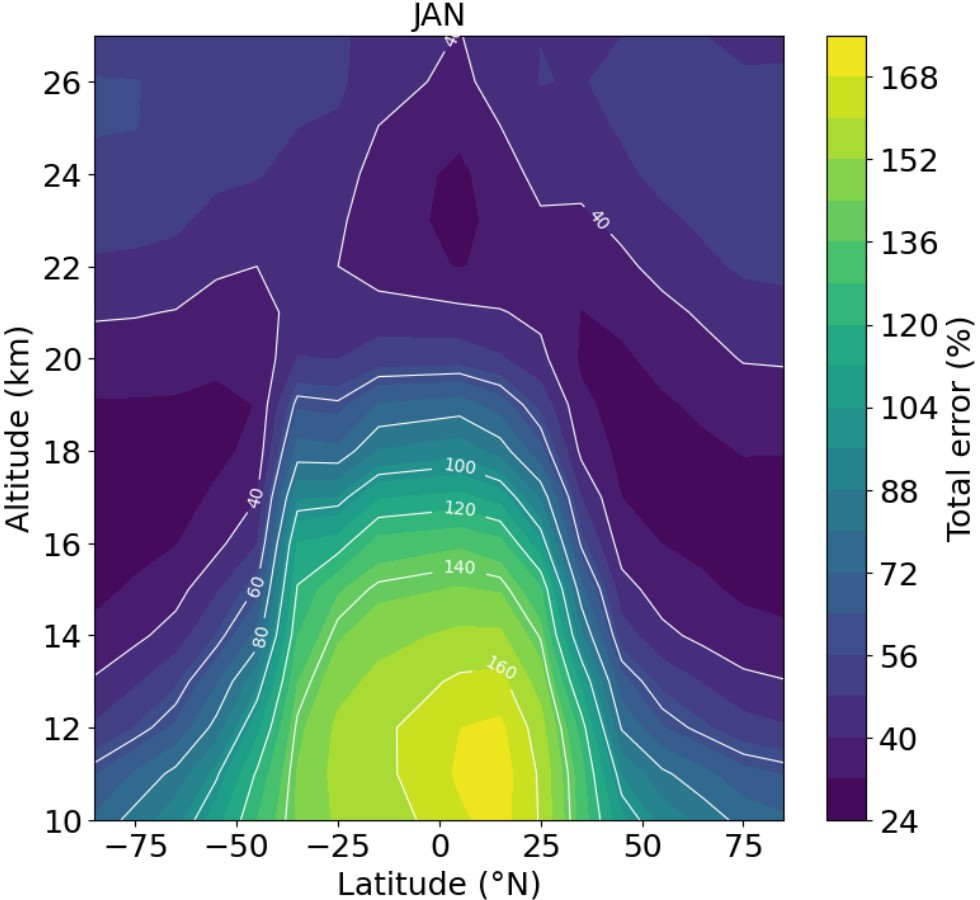

**Figure 7.** Total errors (%) for 0 Tg S/y, i.e. background, and January

Figure 8 illustrates the SAOD (520 nm) from - 55° N to 55° N for 1 Tg S/y in January, including the corresponding total errors, the background case and the true profile (ECHAM simulation results). Analogue to the quasi steady-state phase data, the latitude-dependent tropopause heights from SAGE II data were used for the calculation of the SAOD. While for the quasi steady-state phase data (Fig. 6) the emissions of 1 and 2 Tg S/y in January were observable at all latitudes, this only applies here for ≈ - 10° N to 14° N. However, this is the essential latitude range, since the highest values of the extinction coefficients also occur in this range (Fig. 1, upper panel, subplot (Jan), for 550 nm) and this is also the range in which the SAOD (520 nm) for 1 Tg S/y deviates more clearly from the background case (compare Fig. 8). In addition, Fig. 9 shows the retrieved aerosol extinction profiles at 520 nm for 65° N (upper panel) and 5° N (lower panel), including the total errors, background profiles and true profiles (both at 520 nm). The background profile for 65° N (upper panel) is within the error range for all altitudes considered here, which leads to the conclusion that the emission of 1 Tg S/y cannot be observed at 65° N in January, i.e. the first month of the initial phase. In contrast, this emission can be observed at 5° N (lower panel) at an altitude of ≈ 22 km. The apparent difference regarding the conclusion of the detectability of the geoengineering signal between the aerosol extinction profiles (Fig. 9) and the SAOD (Fig. 8) lies in the method for the calculation of the errors of the SAOD, here the Gaussian error propagation, which leads to a reduction of the SAOD errors.

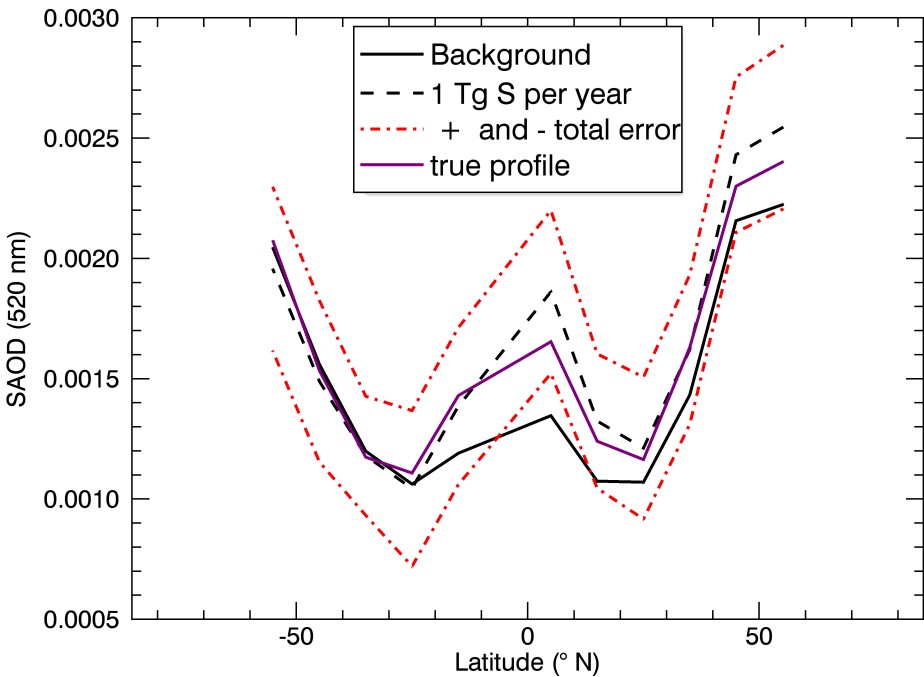

**Figure 8.** SAOD (520 nm) over latitude (° N) for 1 Tg S/y, January in the first year of the initial phase, including the corresponding total errors, the background case and the true profile (ECHAM simulation results).

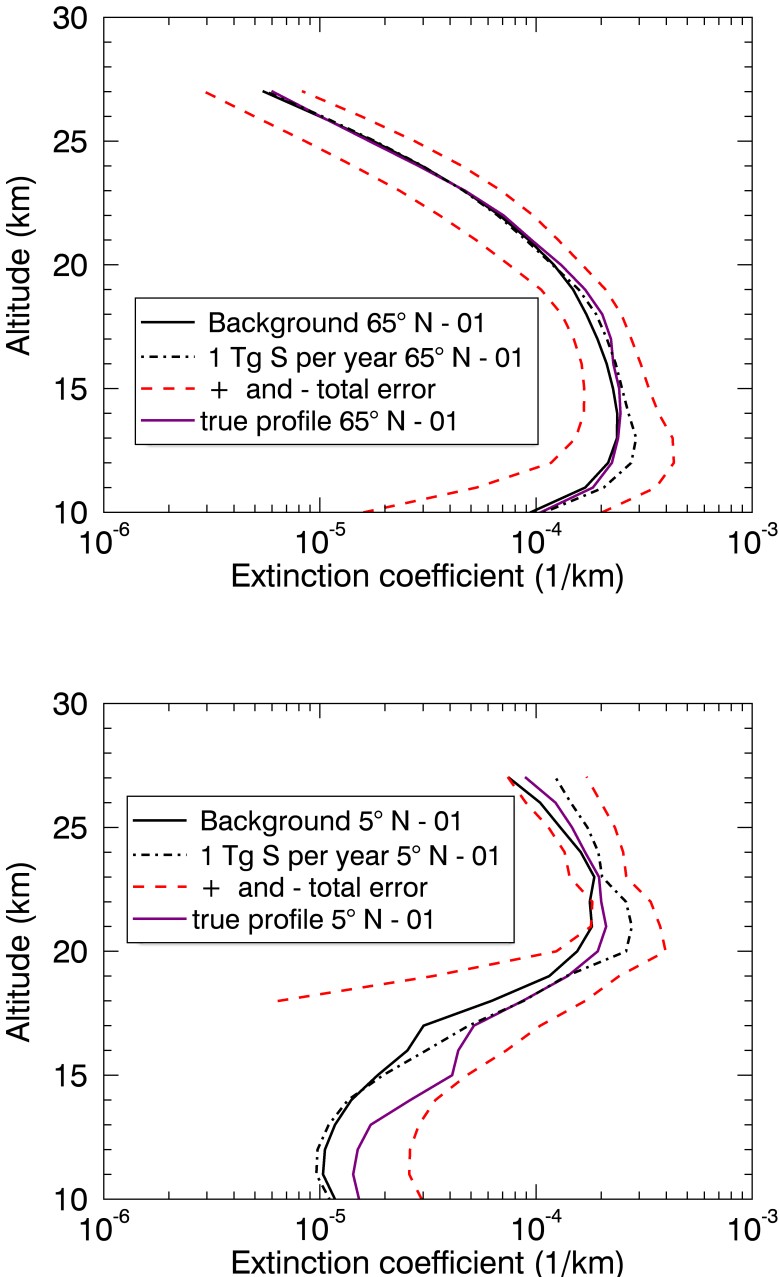

**Figure 9.** Retrieved aerosol extinction profiles at 520 nm for 1 Tg S/y, January, 65° N (upper panel) and 5° N (lower panel), including total errors, background profiles (520 nm) and true profiles (520 nm) (ECHAM simulation results).

In summary, the stratospheric aerosols formed from the emissions of 1 and 2 Tg S/y based on the ECHAM model simulations can be observed for the quasi steady-state phase under the conditions and constraints considered here (see above). Although 1 and 2 Tg S/y do not have a significant climatic effect, these emission rates were chosen to see whether it is possible to detect even these small amounts with a satellite occultation instrument, taking into account an error estimate that is as realistic as possible. The upper limit of the injection amount depends on the specific goal. Depending on the model, 8 to 16 Tg $SO_2$ per year would be required to cool the Earth's surface by 1 degree on a global average (Niemeier, 2023). We assume that with larger injection rates the detectability increases, the aerosol extinction signal becomes larger and the total errors smaller (compare 0 Tg S/y $\rightarrow$ 1 Tg S/y $\rightarrow$ 2 Tg S/y) up to a certain amount, possibly about 20 Tg (as in the case of the Pinatubo eruption, although a volcanic eruption does not represent continuous injections). However, we note that the zero transmittance problem does not mean that solar occultation measurements are entirely useless. They cannot provide aerosol extinction below a certain altitude, but at slightly higher altitudes they will still work and provide information on enhanced aerosols levels. In the case of Pinatubo, SAGE II measurements were always available at altitudes above about 24 km.

At this point, it should be noted that the relatively large errors at low altitudes (compare Figs. 3 and 7) are due to the low extinction coefficients at these altitudes (compare, e.g., Figs. 4 and 5).

Wrana et al., 2021 (Tab. 1) shows the extinction measurement uncertainties at 520 nm averaged from June 2017 to December 2019 at an altitude of 20 km (SAGE III/ISS level 2 solar aerosol product) with a value of 5.66 %. The total errors for 1 and 2 Tg S/y at 20 km are of approximately the same order of magnitude for the northern and southern mid-latitudes. Note that it is difficult to make a comparison, as the present study considers different phases (quasi steady-state phase and initial phase) and different injection rates (1 Tg S/y, 2 Tg S/y and background), which means that only a comparison of the order of magnitude is possible.

The measurement frequencies of, for example, SAGE III/ISS averaged over a given month and latitude for the years 2017 – 2024 are $\approx 20$ measurements for January and $\approx 30$ measurements for July (between 20° S and 20° N). However, the measurement sampling is highly variable and dependent on the month and the respective latitude.

The latitude range investigated here, 85° S – 85° N, is wider than that covered by for example SAGE III/ISS, but the high latitudes were investigated in order to obtain information on the detectability over the largest possible latitude range. The same applies to the latitudinal coverage (due to the actual sampling issues of occultation measurements).

In the analyses so far, the natural variability has not yet been taken into account, which is of course also crucial for drawing conclusions about the detectability of possible geoengineering experiments.

## 3.4 Natural variability

In order to evaluate the magnitude of the natural variability of the SAOD under nearly background conditions, SAGE II data from 2002 – 2004 were used (NASA, 2012). For this purpose, the latitude range 85° S to 85° N (zonal means) and the SAOD at 525 nm were analysed. For different latitudes and months, the mean SAOD and standard deviation were calculated based on the three-year data. To determine the SAOD, the tropopause heights from the SAGE II data were used. The geoengineering signal is considered detectable if the corresponding SAOD is outside the $2\sigma$ range of the natural variability. Figure 10 shows

the mean SAOD at 525 nm over the months from January to December for the SAGE II data from 2002 – 2004, illustrated for 50° S (left panel) and 30° N (right panel). The vertical bars represent the standard deviation of the SAOD values for each month.

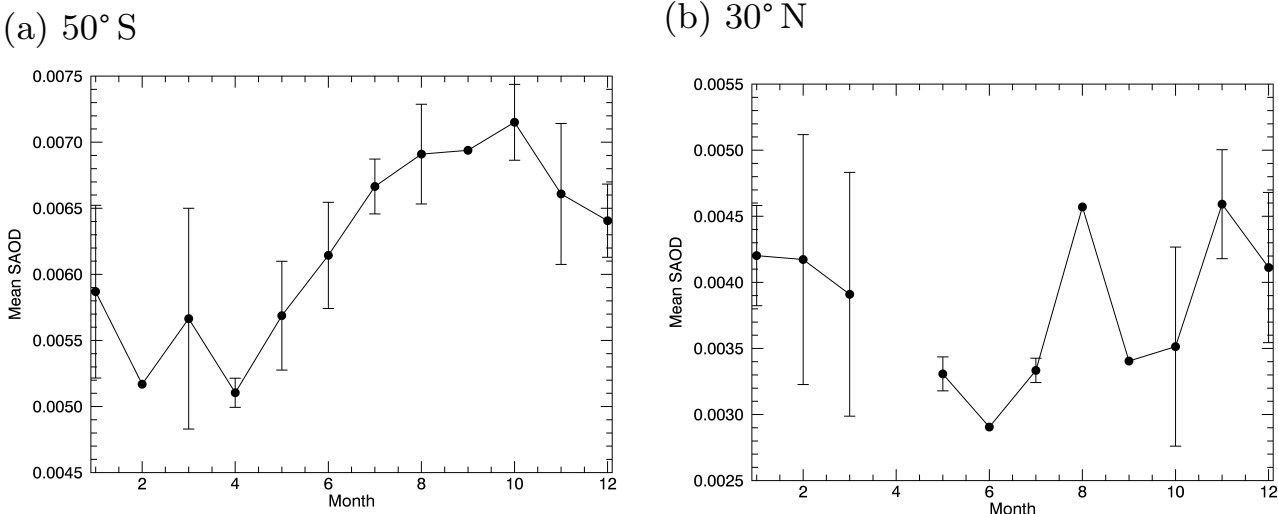

**Figure 10.** Mean SAOD (525 nm) over month for the SAGE II data from 2002 – 2004. The vertical bars indicate the standard deviation of the SAOD values for each month. Illustrated for 50° S (left panel) and 30° N (right panel).

For the quasi steady-state phase and the 1 and 2 Tg S/y (e.g, Fig. 6), the SAOD values lie outside the $2\sigma$ limit, which corresponds to values in the order of $10^{-3}$, while the SAOD values themselves are within the $10^{-2}$ range. This means that the emissions, i.e. the stratospheric aerosols formed, can also be detected taking into account an realistic measure of the natural variability. In contrast, the SAOD values in the first month of the initial phase after the 1 Tg S/y injection are within the range of natural variability in the relevant latitude range (compare e.g, Fig. 8), so at this point it is probably not possible to distinguish the geoengineering signal from the natural variability.

The accuracy of the SAGE II/III data, in this case the aerosol extinction coefficients, can also be affected by instrumental and model-related potential error sources. For example, the pointing accuracy, temperature and pressure profiles, correction of atmospheric refraction and measurement noise (e.g., Damadeo et al., 2013; SAGE III ATBD , 2002).

## 4    Conclusions

Using the ECHAM simulation results and the radiative transfer model SCIATRAN, it is possible to theoretically answer the question of the detectability of possible geoengineering experiments, here the emissions of 1 and 2 Tg S/y, using satellite occultation instruments.

For this, aerosol extinction coefficients at 500 and 550 nm, in an altitude range of 10 to 27 km, simulated with ECHAM were used to calculate the extinction coefficients at 520 nm applying the Ångström parameterisation to simulate transmission values at 520 nm from the perspective of SAGE III/ISS with the radiative transfer model SCIATRAN. These simulated transmission values were then utilised for the retrieval of the stratospheric aerosol profiles with SCIATRAN. A subsequent sensitivity study and error analysis, as well as the consideration of the natural variability using SAGE II data, were applied to answer the question of detectability.

Considering the measurement errors, the natural variability and the assumptions in the simulations with ECHAM and SCI-ATRAN, it is very likely that the formed stratospheric aerosols, from the emissions of 1 and 2 Tg S/y, can be observed in the quasi steady-state phase over all latitudes and months considered here.

In the first month of the initial phase, the signal of the emission of 1 Tg S/y cannot be distinguished from the natural variability, which is why the detection of the formed stratospheric aerosols as geoengineering signal is probably not possible. Accordingly, it can be assumed that smaller emission rates cannot be detected as geoengineering signals during the first month of the geoengineering experiment.

*Code and data availability.* The radiative transfer model SCIATRAN can be downloaded via the following link: https://www.iup.uni-bremen.de/sciatran/. The SAGE II data can be accessed via: https://doi.org/10.5067/ERBS/SAGEII/SOLAR_BINARY_L2-V7.0.

**Appendix A**

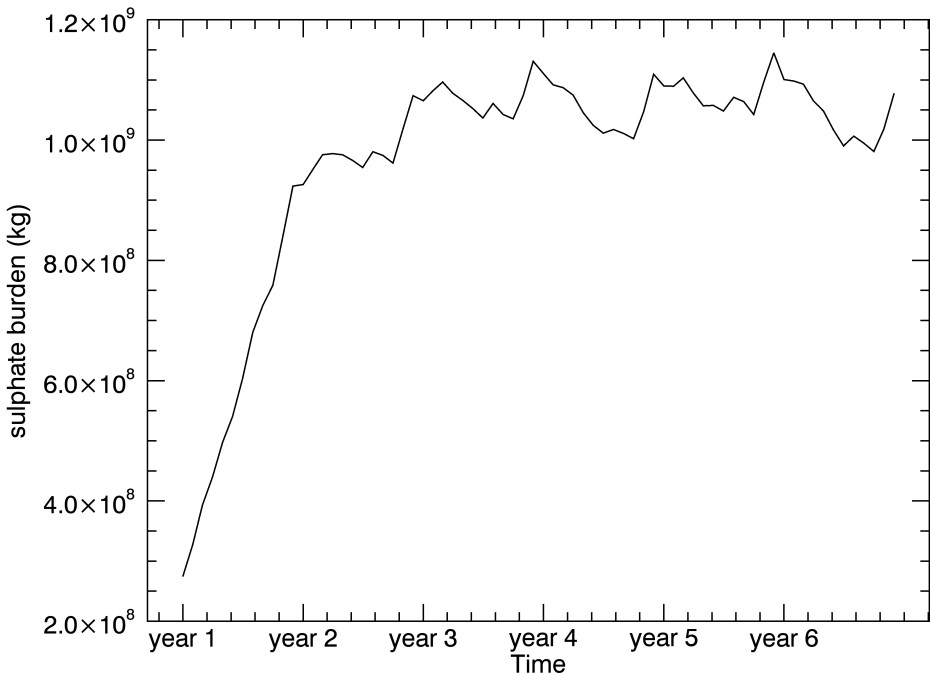

**Figure A1.** Monthly mean sulphate burden in kg over time (2005 – 2010) for 1 Tg S/y, showing the differences between the two-year initial phase and the quasi steady-state phase.

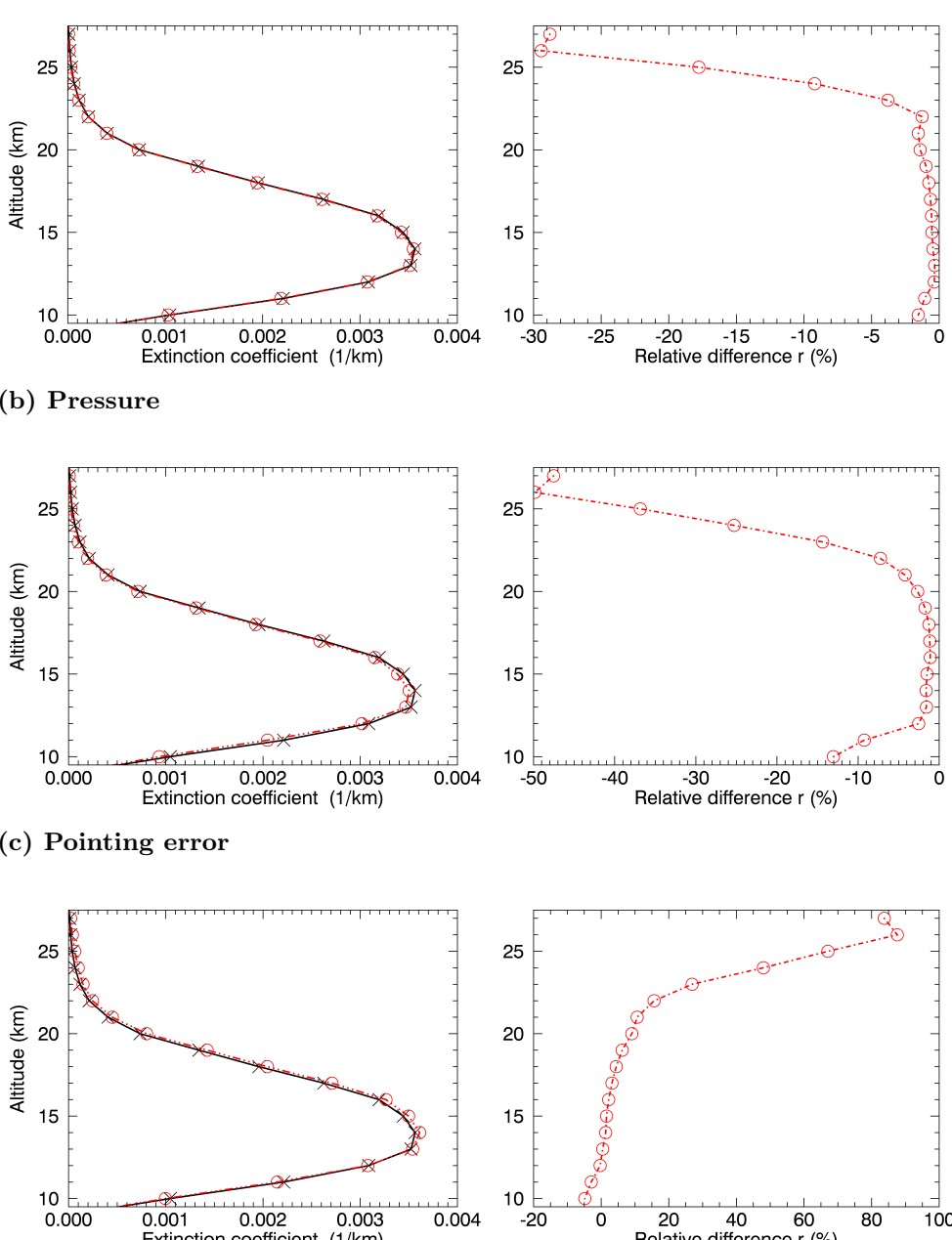

**Figure A2.** Left column: Retrieved aerosol extinction profiles at 520 nm with reference settings (black line) and modified settings (red line) for 1 Tg S/y, 55° N latitude, January based on the ECHAM model simulation results of the quasi steady state phase. Right column: Corresponding relative difference $r$. Both for perturbed (a) Total ozone column, (b) Pressure and (c) shifted tangent height grid.

**(a) Total ozone column**

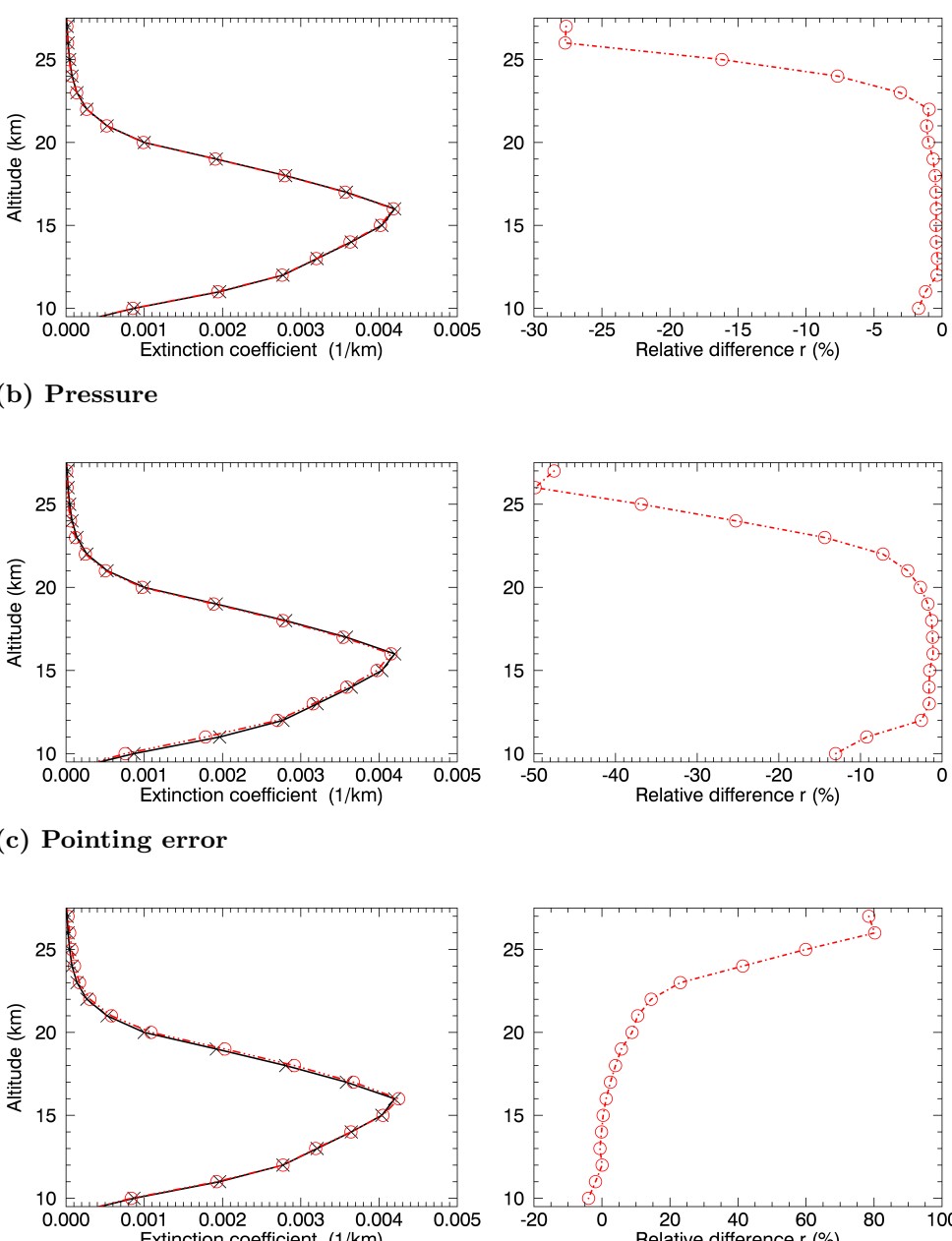

**(b) Pressure**

**(c) Pointing error**

**Figure A3.** Left column: Retrieved aerosol extinction profiles at 520 nm with reference settings (black line) and modified settings (red line) for 1 Tg S/y, 55° S latitude, January based on the ECHAM model simulation results of the quasi steady state phase. Right column: Corresponding relative difference $r$. Both for perturbed (a) Total ozone column, (b) Pressure and (c) shifted tangent height grid.

*Author contributions.* CvS outlined the project and UN performed the simulations using the MAECHAM-HAM model. AL carried out the transmission calculations and retrievals using SCIATRAN with guidance by AR. UN wrote the MAECHAM-HAM methodology subsection. AL wrote an initial version of the paper. All authors discussed, edited and proofread the paper.

*Competing interests.* The authors declare that they have no competing interests.

*Acknowledgements.* We are indebted to the Institute of Environmental Physics of the University of Bremen – particularly to Vladimir Rozanov and John P. Burrows – for access to the SCIATRAN radiative transfer model and Alexei Rozanov for access to the SCIATRAN retrieval algorithm. We thank Christian Löns for providing the SAGE II data. This study was enabled by the collaborations within the DFG research unit Volimpact (FOR 2820, grant no. 398006378).

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
