# Peer review of "Investigating the ability of satellite occultation instruments to monitor possible geoengineering experiments"

_EGUsphere, 2025_

## Author Comment (AC1)

**Replies to comments by Travis N. Knepp**

**Comment:** Hello Anna! Thank you for submitting this paper.

The authors present a non-sophisticated (simplicity is beauty), simple evaluation of a solar occultation instrument's (here, SAGE III/ISS) ability to detect changes in stratospheric aerosol load with a continual injection of 1-2 Tg S/year. This study is elegant in its simplicity. Here, the authors determine that such changes would be detectable. To carry out this study the authors use MAECHAM5-HAM model to generate extinction coefficients at 500 and 550 nm, which were then converted to 520 nm (via Angstrom parameterisation), which was then fed into SCIATRAN to produce transmission data, which was then used to retrieve 520 nm extinction. I believe the accuracy of the authors' conclusions depends on the quality of these models, which I am not qualified to judge.

Overall, this paper is well written and I believe makes an important contribution to the scientific community. It would be interesting to see this study continue to evolve (e.g., if your "natural variability" included pyrocbs and volcanic eruptions (i.e., the stratospheric conditions that SAGE III/ISS has observed under), would you still have the requisite sensitivity to detect changes?; by continually injecting S you are changing the baseline stratospheric aerosol distribution...how does that impact particle growth and radiative transfer after another eruption, etc.), but the authors present an interesting and convincing proof of concept that stands on its own.

I only have minor suggestions for improving the paper.

**Comment:** 1. In Fig. 1 (b), when does injection start? Did it start in January? I apologize if this was mentioned in the text and I missed it, but adding this information to the caption would aid the reader.

**Reply:** Thank you for the comments and the very good suggestions made above, which we also intend to look at in the future.

As described in more detail in the ECHAM subsection, these are continuous emissions, i.e. at each time step, at an altitude of 60 hPa.

**Comment:** 2. For Figures 2, 4, 5, 9, have the authors considered making the x-axis scale logarithmic? Especially in Fig. 4/5 this would help the reader appreciate the magnitude of change from background to enhanced conditions. Currently, I can barely tell that background is different from zero and a log scale would help the reader quantify this.

**Reply:** Thank you, this was changed for the illustrations of the aerosol extinction profiles.

**Comment:** 3. Do you have any information on how particle size may change with these injections? That level of information would be very interesting (at least to me).

Again, well done on the paper. I look forward to seeing this published; best of luck.

**Reply:** Thank you again. We agree that this would be an interesting aspect, but we did not look at this in the context of the aims of this study. We are planning, however, to investigate this aspect in a future study.

---

## Author Response (AR1)

**Replies to comments by reviewer 1**

**Comment:** The manuscript examines the ability of satellite occultation instruments to monitor potential geoengineering experiments using MAECHAM5-HAM simulations and retrievals with the SCIATRAN radiative transfer model. While the model experiments are straightforward and easy to interpret, I have two major concerns regarding the analyses presented in the paper.

**Reply:** We thank the reviewer for his/her constructive and helpful comments. We tried to answer every comment in an appropriate way.

**Comment:** First, the use of data from the initial two years of the simulation raises concerns about whether the model has reached an equilibrium state. It is crucial to ensure that the model's early-phase data are suitable for analysis. Could the authors provide insight into the model's performance during this period and demonstrate that it has stabilized?

**Reply:** We added the following figure to the appendix and explanatory text to the ECHAM section:

[Figure]

**Figure A1.** Monthly mean sulphate burden in kg over time (2005 – 2010) for 1 Tg S/y, showing the differences between the two-year initial phase and the quasi steady-state phase.

The time series of the global sulphate burden shows that the steady-state phase is reached after two years (compare Fig. A1). We used the early phase to include sulphate levels below the steady-state level to see if we could detect sulphate even earlier. At this point, the goal was not to use a stabilized result. The aim was to find a lower threshold at which detection would be possible.

**Comment:** Second, in line 214, the manuscript states, "The errors shown and discussed so far are based on one extinction profile per month." It would be beneficial for the authors to clarify their rationale for this choice and to compare averaged profiles with those currently presented in the manuscript.

**Reply:** Thank you for pointing that out. We were unclear at this point. The aerosol extinction profiles shown from the ECHAM simulations are monthly averages. To avoid further misunderstandings, we have removed the passage.

**Comment:** Beyond these major concerns, the manuscript is well-written and presents a novel approach by using a relatively small sulfur injection to assess the model's response. The study is valuable to the scientific community, and I have provided specific comments below that should be addressed before publication.

Comments:

1. Lines 71-73: How is the model's dynamics treated? Is it prescribed using a climatology? Please elaborate.

**Reply:** MAECHAM is a general circulation model which solves prognostic equations for temperature, surface pressure, vorticity, divergence and phases of water. The model simulates the related dynamical processes and generates the quasi biennial oscillation in the stratosphere. The model is not coupled to an ocean model. Imstead, sea surface temperature is prescribed and set to climatological values (Hurrell et al., 2008), averaged over the AMIP (Atmospheric Model Intercomparison Project) period 1950 to 2000, and does not change due to SAI. No nudging, relaxing the prognostic variables towards an atmospheric reference state to, e.g., ERA5 data, is applied.

We added these additional explanations to the MAECHAM-HAM section.

Hurrell, J. W., Hack, J. J., Shea, D., Caron, J. M., and Rosinski, J.: A New Sea Surface Temperature and Sea Ice Boundary Dataset for the Community Atmosphere Model, J.Climate, 21, 5145–5153, https://doi.org/10.1175/2008JCLI2292.1, 2008

**Comment:** 2. Lines 75-76: Could you clarify whether a single realization or an ensemble is used? If only one realization is run for 15 years, what is the spin-up time? If the first two years and the last three years are analyzed, is the model in equilibrium in the first two years (initial phase)? A time series showing equilibrium would be useful.

**Reply:** We performed a single simulation over several years. The injections for SAI ran for 15 years. For our study, we took three years at the end of these simulations and averaged them over time. This is similar to previous simulations and publications, e.g. Niemeier et al. (2020), Weisenstein et al. (2022), where we also used a three-year average. The sulphate burden for an injection of $1 \, \text{Tg/y}$ stabilises after two years (compare Fig. A1). See our previous answer above regarding the early phase. We added this explanations to the lines 75-76.

Niemeier, U., Richter, J. H., and Tilmes, S.: Differing responses of the quasi-biennial oscillation to artificial SO2 injections in two global models, Atmos. Chem. Phys., 20, 8975–8987, `https://doi.org/10.5194/acp-20-8975-2020`, 2020.

Weisenstein, D. K., Visioni, D., Franke, H., Niemeier, U., Vattioni, S., Chiodo, G., Peter, T., and

Keith, D. W.: An interactive stratospheric aerosol model intercomparison of solar geoengineering by stratospheric injection of SO2 or accumulation-mode sulfuric acid aerosols, Atmos. Chem. Phys., 22, 2955–2973, https://doi.org/10.5194/acp-22-2955-2022, 2022.

**Comment:** 3. Line 96: Correct the typo "trough" to "through."

**Reply:** Changed.

**Comment:** 4. Lines 97-98: Are these climatological values equivalent to those used in the model if the dynamics are prescribed?

**Reply:** As described in the corresponding section of the preprint, we use the vertical profiles of pressure, temperature and trace gases from a climatological database implemented in SCIATRAN, which is based on a 3-D chemical transport model. These values will not be identical to those in ECHAM.
Possible errors in the derived aerosol extinction profiles due to inaccurate knowledge or assumptions in e.g. pressure and temperature are part of the sensitivity study we have performed.

**Comment:** 5. Lines 120-121: Is this statement based on observations or model experiments? Have you conducted a no-sulfur (0 Tg S/year) experiment to establish a background aerosol extinction coefficient? Please see my comment for Figure 7 (lines 187-188).

**Reply:** The lines say: 'The retrieval was restricted to the altitude range of the aerosol extinction coefficients provided (see Sect. 2.1), i.e. from 10 to 27 km. Background aerosol coefficient profiles at 520 nm corresponding to the latitude and month were used as a priori information.'
The altitude restriction is based on the altitude range of the given aerosol extinction profiles from ECHAM, which is why we have referred to the ECHAM section (Section 2.1). The background profiles originate from the ECHAM simulations as the following line in the preprint says: 'These profiles originate from the ECHAM5-HAM model simulations (compare Sect. 2.1)'. (lines 121 - 122).
The background profiles correspond to 0 Tg S/y. To prevent further misunderstandings, we have added this information to the sentences mentioned here.

**Comment:** 6. Line 120: Insert "extinction" between "aerosol" and "coefficient" for clarity.

**Reply:** Changed.

**Comment:** 7. Line 124: Could you please define "quasi-steady state" and "initial phase" more explicitly, especially in relation to model equilibrium.

**Reply:** We added additional explanations (see above).

**Comment:** 8. Lines 127-130: What is the reasoning behind showing these figures? Are they meant to highlight differences between the initial and quasi-steady-state phases? If so, please clarify.

**Reply:** Yes, you are right. We have added this figure to the preprint to show the differences between the initial phase and the quasi steady-state phase.
We have now added this explanation.

**Comment:** 9. Lines 140-145: The sensitivity analysis is appreciated. Could you elaborate on why differences are larger below 15 km and improve above that altitude? Please do the same for the appendix figures.

**Reply:** Thank you for the question. We have addressed this point in the discussions: 'At this point, it should be noted that the relatively large errors at low altitudes (compare Figs. 3 and 7) are due to the low extinction coefficients at these altitudes (compare, e.g., Figs. 4 and 5).' (lines 212 – 213).

**Comment:** 10. Lines 149-155: Why return to the initial-phase data here? If the quasi-steady-state phase is used for analysis, maintaining consistency would be preferable. I doubt if the initial phase data and their analyses are relevant as the model may not have attained equilibrium state by then (first two years).

**Reply:** Thank you for the comment. Please see our previous detailed answers above (replies to the major concerns).

**Comment:** 11. First line of section 3.2: Line 149 mentions that retrievals are based on initial-phase data, but this section focuses on quasi-steady-state data. Why mix the two?

**Reply:** We think there is a misunderstanding here. Line 149 is not in the subsection on the quasi steady-state phase, but still in the general section on results and discussions (in subsection 3.1 Sensitivity study), which is why both phases are addressed in this section.

**Comment:** 12. Third line of section 3.2: Could you explain why total errors for 1 Tg S/y are greater than for 2 Tg S/y in Figure 3? This seems counterintuitive.

**Reply:** We have now added the explanations to line 158. It now says: 'Overall, the total errors for 1 Tg S/y are greater than for 2 Tg S/y, which is consistent with the expectations, as the signal is stronger at a higher injection rate such as 2 Tg S/y and the total errors are therefore smaller.'

**Comment:** 13. Line 162: Please clarify the term "true profiles." Does this refer to model simulation results?

**Reply:** Yes, you are right. The information is in lines 162 – 163. The entire passage says: 'The following Figs. 4 and 5 show the retrieved aerosol extinction profiles at 520 nm (black dashed lines)

for 1 Tg S/y (Fig. 4) and 2 Tg S/y (Fig. 5), both for January (left column) and July (right column) including the corresponding total errors (red dashed lines), the background profiles (black solid lines) and true profiles (purple solid lines) both at 520 nm (ECHAM5-HAM model simulation results) for 65° N (upper panels), 15° N (middle panels) and 65° S (lower panels).'.

To prevent further misunderstandings, we added this information to other passages in the preprint when the term "true profile" is mentioned.

**Comment:** 14. Line 175: Please replace ± 55°N with 55S-55N or something similar.

**Reply:** Replaced with - 55 to 55° N .

**Comment:** 15. Line 178: Do you have corresponding model simulation values for Figures 4 and 5? If so, please include them.

**Reply:** This refers to minor point 13 (see above). The 'true profiles' are the results of the ECHAM simulations, as explained above. This means that the ECHAM simulations are shown in the figures mentioned. We added this information in the corresponding text and captions.

**Comment:** 16. Lines 180-181: Adding model-simulated SAOD using the 55S-55N latitude band would strengthen comparisons. Can you provide insight into the causes of maxima and minima in these figures?

**Reply:** Thank you for the suggestion. We added the model-simulated SAOD and added the following explanations to the text: "The maximum in SAOD near the equator is due to the fact that 1 or 2 Tg S/y are continuously injected in this region (compare Sect. 2.1). The minima of the SAOD in the subtropics and maxima at mid and high latitudes are due to the latitude dependence of the tropopause height, which is higher in the tropics (about 18 km) than at the mid and high latitudes (about 8 km).".

**Comment:** 17. Lines 187-188: Section 3.2 is based on quasi-steady-state data, but here total errors seem to be derived from the initial phase. This is confusing—please clarify. Also, if you are analyzing a background case (0 Tg S/y) from the first month of the initial phase, a table in Section 2.1 listing all model simulations (0 Tg S/y, 1 Tg S/y, and 2 Tg S/y) would improve clarity.

**Reply:** We think there is a misunderstanding. Lines 187 – 188 are in subsection 3.3 Initial phase data (and not subsection 3.2 quasi steady-state), which is why the first sentence in this subsection refers to the initial phase.

**Comment:** 18. Lines 190-192: Could you please elaborate a bit more on as to why the total errors would be larger for Figure 7 compared to Figure 3?

**Reply:** Thank you for pointing this out. In the course of answering point 12 (see above), we added

additional explanations.

**Comment:** 19. Lines 206-207: What is the rationale for using only the first month of the initial phase? How is that related to detectability of satellite occultation instruments? Would the model have reached equilibrium by then? Please see my comments above for line 75.

**Reply:** Thank you for the comment. Please see our previous detailed answers above (replies to the major concerns).

**Comment:** 20. Line 214: How was the selection made to use one extinction profile per month for Figures 2, 4, 5 and 9? It would be beneficial for the authors to clarify their rationale for this choice and to compare averaged profiles with those currently presented in the manuscript. Also, please consider showing model SAOD values in Figures.

**Reply:** As explained above (major point 2), there is a misunderstanding here due to our wording, which we have corrected (see above). The data from the ECHAM simulations are monthly averages.

**Comment:** 21. Lines 219-220: SAGE III/ISS measurements were not available before 2017. Please correct this.

**Reply:** Thank you for pointing this out, we corrected that. It now says: "2017 – 2024".

**Comment:** 22. Lines 236-237: To highlight differences, please consider overlaying SAOD values from Figure 6 onto Figure 10 (e.g., plotting SAOD for 1 and 2 Tg cases at 50S and 30N from January to December).

**Reply:** Thank you for the suggestion. We tested this, but decided against it, as the graphical representation as a whole is less clear (rather messy) and the informative value at 'a glance' is lost. Nevertheless, we understand that such a visualisation can theoretically be practical for displaying differences.

**Comment:** 23. Lines 258-260: The discrepancy discussed here may result from the model's non-equilibrium state during the initial phase. To ensure consistency, it would be preferable for the analyses to focus on data from the model's equilibrium (quasi steady) state.

**Reply:** Thank your for the constructive comment. Please see the detailed answers to the questions about the initial phase above.

**Replies to comments by reviewer 2**

**Comment:** Dear Editor, dear Authors,

The manuscript "Investigating the ability of satellite occultation instruments to monitor possible geoengineering experiments" by Lange et al. aims at analysing the expected capability of solar occultation instruments like SAGE III to detect and characterise stratospheric aerosol injection (SAI) geoengineering interventions. The topic of the manuscript is potentially important and relevant for a readership of satellite instrument and data scientists. Despite this, I have a few major concerns (especially Major Comment 1 below) about the manuscript and the inherent analyses that must be addressed before the manuscript can be considered for publication, in my opinion. I also have minor and specific comments, listed in the following. Please address the following major and minor comments and I will be willing to further evaluate a new manuscript version.

Regards.

**Reply:** We thank the reviewer for his/her constructive and helpful comments. We tried to answer every comment in an appropriate way.

**Comment:** Major Comments:

1. The overarching motivation of this work is to assess the detectability of SAI interventions with SAGE III-like instruments. To do so, two geoengineering modelling experiments have been used as a pseudo-reality scenario, simulating SAI with injections of 1 or 2 Tg S/year. Now, different moderate stratospheric eruptions occurred in the last few years, e.g. Raikoke 2019, Hunga 2022, and more in the previous after-Pinatubo years, each with SO2 injection of that order of magnitude (1-2 Tg SO2 or less → 0.5-1.0 Tg S or less). Many scientific works exist that show detection, tracking and quantitative characterisation of these plumes with SAGE III/ISS, e.g. More quantitatively from an aerosol extinction point of view, the perturbation in the aerosol extinction for this geoengineering experiment and study is of the order of 10-3 km-1 at 550 nm. This is quite comparable to perturbations of recent moderate eruptions like Raikoke, which sulphate aerosol perturbation was successfully detected studied and quantitatively characterised with SAGE III/ISS (e.g. Kloss et al., 2021, https://acp.copernicus.org/articles/21/535/2021/). So, even without this study, one could affirm that, yes, geoengineering plumes resulting from injection of 1 (or even less) Tg S and producing aerosol extinction perturbation of the order of 10-3 km-1 at 550 nm are observable and measurable with SAGE III. Thus, it is mandatory to clarify the scopes of this manuscript, restructure the manuscript accordingly and possibly change the title.

**Reply:** Thank you for this constructive comment. We agree that moderate volcanic eruptions such as Raikoke (2019) and Hunga (2022) have resulted in sulphur injections comparable in magnitude to those used in our simulations. However, the key difference between a volcanic eruption, such as Raikoke, and the emissions in our study is hat the emissions here are continuous, $1\,\text{Tg/y}$ was continuously injected at every model time step, which results in a much lower sulfate injection per time compared to a volcanic eruption with the same injected amount (in the first month of the initial phase, only $1/12\,\text{Tg S/y}$ were emitted)).

To make it clearer we added this information to the introduction section.

**Comment:** 2. The readership of interest for this kind of manuscript is more AMT that ACP, as it is very centred on instrumental aspects. Thus, I propose to transfer the manuscript to that journal.

**Reply:** Thank you for the suggestion. We think that the topic of our paper is exactly within the scope of ACP. In addition, the preprint was approved by the editor for the peer review process in ACP, which is why we want to keep ACP as a journal.

**Comment:** 3. I don't really understand the background idea behind Sect. 3.3. Why is the error analysis performed for background case to study the initial phase of the perturbed case? Why not just making an error analysis of the initial phase itself instead? Why you conclude that "The background profile for 65° N (upper panel) is within the error range for all altitudes considered here, which leads to the conclusion that the emission of 1 Tg S/y cannot be observed at 65° N in January, i.e. the first month of the initial phase."? And why do you conclude that "the stratospheric aerosols formed in the first month of the initial phase can only be detected with satellite occultation instruments in a limited latitude range from – 10° to 14° N." All this is unclear to me and should be clarified.

**Reply:** Regarding point 1: For the investigation of the initial phase, we used the errors of the background case based on the assumption that these errors are larger. We added this fact to the corresponding section.

Regarding point 2: Thank you for bringing this up, we understand that the wording can sound a little confusing. The following sentence: "The background profile for 65° N (upper panel) is within the error range for all altitudes considered here, which leads to the conclusion that the emission of 1 Tg S/y cannot be observed at 65° N in January, i.e. the first month of the initial phase." refers to Fig. 9 showing the retrieved aerosol extinction profiles at 520 nm for 1 Tg S/y, January, 65° N (upper panel) and 5° N (lower panel), including total errors, background profiles (520 nm) and true profiles (520 nm). The sentence refers to the detectability of a possible geoengineering signal.

The other sentence: "The stratospheric aerosols formed in the first month of the initial phase can only be detected with satellite occultation instruments in a limited latitude range from – 10° to 14° N." is part of the conclusions and refers to Fig. 8 showing the SAOD (520 nm) over latitude (° N) for 1 Tg S/y, January in the first year of the initial phase, including the corresponding total errors and the background case, and discussing the detectability.

We understand that the wording of the statements can be confusing. We therefore removed the last statement from the conclusions. The mentioned signal cannot be distinguished from the natural variability, as also discussed at the corresponding section (subsection 3.4, ll. 239 – 241) and in the conclusions (ll. 258 – 260). The corresponding passage in the conclusions now says: 'In the first month of the initial phase, the signal of the emission of 1 Tg S/y cannot be distinguished from the natural variability, which is why the detection of the formed stratospheric aerosols as geoengineering signal is probably not possible.'

**Comment:** 4. The quality of the text and clarity must be improved throughout the whole manuscript, more in the Specific Comments.

Minor Comments:

1. L14: "and increase" → "by increasing"

**Reply:** Changed.

**Comment:** 2. L15: "SAI" is rather "stratospheric aerosol injection"

**Reply:** Changed.

**Comment:** 3. L16 and Introduction in general: please consider citing more recent literature, e.g. https://www.frontiersin.org/journals/climate/articles/10.3389/fclim.2025.1507479/full

**Reply:** We added:
- Haywood et al., 2025 (a)
- Janssens et al., 2020 (b)
- Weisenstein et al., 2022 (c)
- Quaglia et al., 2022 (d)
(a) Haywood, J. M., Boucher, O., Lennard, C., Storelvmo, T., Tilmes, S., and Visioni, D.: World Climate Research Programme lighthouse activity: an assessment of major research gaps in solar radiation modification research. Frontiers in Climate, 7, 1507479, 2025.
(b) Janssens, M., de Vries, I. E., and Hulshoff, S. J.: A specialised delivery system for stratospheric sulphate aerosols: design and operation. Clim. Chang. 162, 67–85. doi: 10.1007/s10584-020-02740-3, 2020.
(c) Weisenstein, D. K., Visioni, D., Franke, H., Niemeier, U., Vattioni, S., Chiodo, G., et al.: An interactive stratospheric aerosol model intercomparison of solar geoengineering by stratospheric injection of SO2 or accumulation-mode sulfuric acid aerosols. Atmos. Chem. Phys. 22, 2955–2973. doi: 10.5194/acp-22-2955-2022, 2022.
(d) Quaglia, I., Visioni, D., Pitari, G., and Kravitz, B.: An approach to sulfate geoengineering with surface emissions of carbonyl sulfide. Atmos. Chem. Phys. 22, 5757–5773. doi: 10.5194/acp-22-5757-2022, 2022.

**Comment:** 4. L17: "sulphur" → "sulphur dioxide"

**Reply:** Changed.

**Comment:** 5. L19: "forcing" → "radiative forcing"

**Reply:** Changed.

**Comment:** 6. L20-21: "Simulations...W/m2" after how much time of 10 Tg S/y injections?

**Reply:** SO2 was injected continuously at a height of 60 hPa (about 19 km) (Niemeier and Timmreck,

2015). We added this fact to the lines 20 – 21.

Referring to: Niemeier, U. and Timmreck, C.: What is the limit of climate engineering by stratospheric injection of SO2?, Atmos. Chem. Phys., 15, 9129–9141, https://doi.org/10.5194/acp-15-9129-2015, 2015.

**Comment:** 7. L26-27: I don't get the logic here. Why now it is mentioned that it will be important to observe "small amounts of *sulphate aerosols" while before "large eruptions" where mentioned?

**Reply:** Thank you for the comment. L. 26-27 refer to the company like "Make Sunsets", which promotes small injection amounts. This also highlights the importance of investigating even small injection amounts, based on the assumption that if these smaller amounts of sulphur are detectable, larger amounts will also be detectable.

The effects of large volcanic eruptions, especially the possible decrease in global mean surface temperature, serve as a model for SAI, which is why large volcanic eruptions were mentioned above this sentence (L. 26-27).

**Comment:** 8. L28 "SAGE" acronym not defined here. Please check if all acronyms are defined.

**Reply:** Checked and changed.

**Comment:** 9. L29: "active" sounds odd here (SAGE is "passive" so this word can be taken in a different sense), also please mention "*solar occultation" and also why "one of the..."? Are there other solar occultation missions operational at the moment?

**Reply:** Thank you for the comment, we changed that. The sentence now says: 'The SAGE III/ISS (Stratospheric Aerosol and Gas Experiment III) instrument, mounted on the International Space Station (ISS), is a currently operating satellite solar occultation instrument.'.

**Comment:** 10. L32; really there is nothing more recent than McCormick et al., 1979?

**Reply:** We have included this reference because it is officially stated in the NASA User's guide for SAGE III/ISS (National Aeronautics and Space Administration (NASA): Stratospheric Aerosol and Gas Experiment on the International Space Station(SAGE III/ISS). Data Products User's Guide Version 5.21, Langley Research Center, 2022.). We are aware that there are more recent publications on this, so we have rephrased it as follows: "A description of the solar occultation measurement technique can be found in, e.g., McCormick et al. (1979)."

**Comment:** 11. L35: is 1-2 Tg S/y really a "small" injection? This sound like over a Raikoke-type eruption per year, so much larger than natural occurrence. In my opinion, this cannot be called "small". Please rephrase.

**Reply:** Thank you for the comment. We agree that $1 – 2$ Tg S/y are comparatively moderate

amounts for volcanic eruptions. However, for possible geoengineering experiments, $1 - 2$ Tg S/y are comparatively small amounts. Depending on the model, 8 to 16 Tg SO2 per year would be required to cool the Earth's surface by 1 degree on a global average (Niemeier, 2023). For comparison, the eruption of Mt Pinatubo in 1991, released $\approx 20$ Tg SO2 into the stratosphere.

Nevertheless, we have reworded this and added the phrase 'in the context of possible geoengineering experiments'.

Referring to: Niemeier, Ulrike. (2023). Eine künstliche stratosphärische Schwefelschicht: Der einfache Ausweg aus dem Klimaproblem? (Version 1. Aufl.). In WARNSIGNAL-KLIMA: Hilft Technik gegen die Erderwärmung ? Climate Engineering in der Diskussion (pp. 243–249). Hamburg, Germany: Wissenschaftliche Auswertungen in Kooperation mit GEO Magazin-Hamburg. `http://doi.org/10.25592/uhhfdm.12856`

**Comment:** 12. -0.3 to -0.6 W/m2 is 10-20% the total greenhouse gases radiative forcing since preindustrial era, and up to twice the 10-years greenhouse gases radiative forcing; so why should this not be considered a significant radiative forcing?

**Reply:** Yes, you are right, but in the context of possible geoengineering experiments (with relevant climate effects), are these comparatively small radiative forcings (as explained in the preprint). We added the phrase 'in the context of possible geoengineering experiments'.

**Comment:** 13. L39: again, these are not small amounts of SO2 injections but rather > 1 Raikoke per year...

**Reply:** We added the phrase 'in the context of possible geoengineering experiments'.

**Comment:** 14. L55: "truncation at wavenumber 63" what does it mean?

**Reply:** MAECHAM is a spectral model and the truncation number describes the horizontal resolution of the model. We changed the order of the sentence in the model description to make this a bit clearer: MAECHAM was applied with a grid size of about 1.8° x 1.8°, more specific the spectral truncation at wavenumber 63 (T63), and 95 vertical layers up to 0.01 hPa (about 80 km).

**Comment:** 15. L66-67: What are the optical properties of sulphate aerosol in input to the radiation scheme? Are they calculated using an online Mie code and using the size distributions obtained with the model?

**Reply:** The solar radiation scheme in ECHAM has 4 spectral bands, 1 for the visible and ultra-violet, and 3 for the near-infrared. The long-wave radiation scheme has 16 spectral bands (see Stier et al, 2003). Values for single scattering albedo and asymmetry factor are taken from a look-up table, pre calculated with Mie calculations, to save computation time.

Stier, P., Feichter, J., Kinne, S., Kloster, S., Vignati, E., Wilson, J., Ganzeveld, L., Tegen, I., Werner,

M., Balkanski, Y., Schulz, M., Boucher, O., Minikin, A., and Petzold, A.: The aerosol-climate model ECHAM5-HAM, Atmos. Chem. Phys., 5, 1125–1156, https://doi.org/10.5194/acp-5-1125-2005, 2005.

**Comment:** 16. L74: is sulphur injected as SO2? Which is the rate of SO2 injection - one shot per year or different injections finally cumulating to 1 or 2 Tg/y?

**Reply:** As described in the ECHAM subsection (2.1), SO2 was injected continuously at an altitude of 60 hPa.

**Comment:** 17. Section 2.2: I don't think that SCIATRAN is properly introduced (what is it? how it works? relevant publications, etc)

**Reply:** We added the following to the section: "The SCIATRAN radiative transfer model was developed by the Institute of Environmental Physics at the University of Bremen, Germany (Rozanov et al., 2014). SCIATRAN was originally designed for satellite-based data retrieval. More information can be found at `https://www.iup.uni-bremen.de/sciatran/` (last access: 30 April 2025)."

**Comment:** 18. Eqs 1 and 2: can you mention here a typical obtained value for the Angström exponent?

**Reply:** The exact values depend on the e.g. considered month, latitude and injection scenario. For example, for January, 45°N and background (0 Tg S/y), the values are between $\approx$ 1.0 and 2.2, depending on the altitude.
We added this information to the "Transmission calculations" subsection.

**Comment:** 19. L104-105: "which is not publicly available": OK but is there at least a previous publication where it is used or this is the first one?

**Reply:** Yes, there are other publications in which the SCIATRAN retrieval algorithm was used. We cited the paper that describes the SCIATRAN retrieval algorithm (Rozanov et al., 2011).
Rozanov, A., Kühl, S., Doicu, A., McLinden, C., Puķīte, J., Bovensmann, H., Burrows, J. P., Deutschmann, T., Dorf, M., Goutail, F., Grunow, K., Hendrick, F., von Hobe, M., Hrechanyy, S., Lichtenberg, G., Pfeilsticker, K., Pommereau, J. P., Van Roozendael, M., Stroh, F., and Wagner, T.: BrO vertical distributions from SCIAMACHY limb measurements: comparison of algorithms and retrieval results, Atmos. Meas. Tech., 4, 1319–1359, https://doi.org/10.5194/amt-4-1319-2011, 2011.

**Comment:** 20. L110: "The apriori state vector is kept constant" constant with respect to what? And constant at which value?

**Reply:** As described in the preprint (l. 120 – 121), the background aerosol extinction profiles at 520 nm (from the ECHAM simulations) are used as a priori information (for the corresponding month

and latitude).

$x_a$ is kept constant over the iteration steps. We have added this information to line 110.

**Comment:** 21. 1: I'm not sure that the meaning of this figure is clear, especially because of the different color scales. Basically, why this figure is shown? To show how much the extinction coefficient is perturbed by SAI in the simulations? In this latter case, please use the same color scale for both panels.

**Reply:** In the process of addressing the comments from reviewer 1, we added explanatory information to Figure 1. The figure aims to visualise the differences between the initial phase and the quasi steady-state phase. We added this information to the text.

Due to the large range of values, it is unfortunately not possible to use the same colour bars for each panel.

**Comment:** 22. L133-138: how are these modified settings chosen? It would be more significant to take variations that are comparable with uncertainties of O3, T, P from other (best available) measurements, and of realistic pointing errors.

**Reply:** Thank you for the comment. We think that the stated uncertainties are realistic and added the relevant literature to Table 3. For the temperature and pressure uncertainties: e.g.: Nowlan et al. (2007) and Langland et al. (2008). For the total ozone column: e.g.: Garane et al. (2019) and the pointing error: e.g.: Bramstedt et al. (2012).

Nowlan, C., McElroy, C., and Drummond, J.: Measurements of the O2 A-and B-bands for determining temperature and pressure profiles from ACE–MAESTRO: Forward model and retrieval algorithm, Journal of Quantitative Spectroscopy and Radiative Transfer, 108, 371–388, 2007.

Langland, R. H., Maue, R. N., and Bishop, C. H.: Uncertainty in atmospheric temperature analyses, Tellus A: Dynamic Meteorology and Oceanography, 60, 598–603, 2008.

Garane, K., Koukouli, M.-E., Verhoelst, T., Lerot, C., Heue, K.-P., Fioletov, V., Balis, D., Bais, A., Bazureau, A., Dehn, A., Goutail, F., Granville, J., Griffin, D., Hubert, D., Keppens, A., Lambert, J.-C., Loyola, D., McLinden, C., Pazmino, A., Pommereau, J.-P., Redondas, A., Romahn, F., Valks, P., Van Roozendael, M., Xu, J., Zehner, C., Zerefos, C., and Zimmer, W.: TROPOMI/S5P total ozone column data: global ground-based validation and consistency with other satellite missions, Atmos. Meas. Tech., 12, 5263–5287, https://doi.org/10.5194/amt-12-5263-2019, 2019.

Bramstedt, K., Noël, S., Bovensmann, H., Gottwald, M., and Burrows, J. P.: Precise pointing knowledge for SCIAMACHY solar occultation measurements, Atmos. Meas. Tech., 5, 2867–2880, https://doi.org/10.5194/amt-5-2867-2012, 2012.

**Comment:** 23. Eq. 6 is formally wrong because this is not expressed in %, as in the corresponding figure.

**Reply:** We corrected Eq. 6 to:

$$r = \frac{x - \text{ref}}{\text{ref}} \cdot 100\%$$

**Comment:** 24. Fig.2: what does it mean, in the caption, the mention to "5°N"?

**Reply:** 5°N refers to the latitude for which the figure is shown (not all latitudes investigated can be shown, some others can be found in the appendix of the preprint). We have now specified this to make it clearer. The caption now says: "Left column: Retrieved aerosol extinction profiles at 520 nm with reference settings (black line) and modified settings (red line) for 1 Tg S/y, 5° N latitude, January based on the ECHAM model simulation results of the quasi steady state phase. Right column: Corresponding relative difference $r$. Both for perturbed (a) Total ozone column, (b) Pressure and (c) shifted tangent height grid." We also specified the captions of the figures in the appendix.

**Comment:** 25. I don't quite get the sense of sentence at L149-150.

**Reply:** This section describes the procedure for the sensitivity study. L. 149 - 150 provide information on the handling of the data from the initial phase, which we believe is important for understanding the following.

**Comment:** 26. Isn't it rather the square root of Eq. 7 what is actually called, before and after in the text, the "total error"?

**Reply:** Yes, you are right. To clarify this, we added the following to the explanation of Eq. 7: "The term "total error" used in the following refers to the square root of Eq. 7.".

**Comment:** 27. Section 3.2: can you quantify the typical total error in the stratosphere (the altitude of SAI injection)?

**Reply:** The total errors (in %) for the data of the quasi steady-state phase at the altitude of the SAI injection (here 60 hPa ≈ 19 km) depend on the emission rate, month and latitude:
1 Tg S/y:
January: 10 % (highest value) (85° N), 3 % (lowest value) (25° N, 25° S, 35° S)
July: 47 % (highest value) (85° S), 3 % (lowest value) (35° S)
2 Tg S/y:
January: 8 % (highest value) (85° N), 3 % (lowest value) (25° N, 25° S, 35° S)
July: 47 % (highest value) (85° S), 3 % (lowest value) (35°N, 25° S)

We added this to the quasi steady-state phase data subsection.

**Comment:** 28. L160: Remove "The following"

**Reply:** Removed.

**Comment:** 29. L161: "January and July" of which year?

**Reply:** As described in the methodology section (Section 2.1, l. 78), the data for the quasi steady-state phase is an average over three years (here the years 12 to 15). To make it clearer, we added this information to l. 161.

**Comment:** 30. Fig. 4: purple lines are difficult to see: why not another color, e.g. green?

**Reply:** Our choice of colours in the preprint was based on the ACP guidelines for illustrations, which also specify colour-blind friendliness. That's why we decided against the 'rainbow colours'.

**Comment:** 31. L164-166 and L180-185 (also Major Comment 1): the fact that these SAI interventions are visible with SAGE III is not really a surprise because these are not "small" injections but rather the same level of moderate eruptions, like Raikoke 2019, Hunga 2022, etc, which have been already successfully observed with SAGE III/ISS

**Reply:** See detailed comment above. But the main difference between the volcanic eruptions mentioned and the emission scenarios here is that the emissions here are continuous (which is not the case for a volcanic eruption).

**Comment:** 32. Section 3.3: see Major Comment 3

**Reply:** See detailed reply above.

**Comment:** 33. L210: "Although...effect" see my previous comment on this

**Reply:** See detailed reply above.

**Comment:** 34. L236-237: why not showing the geoengineering signal here (e.g. in Fig. 10)?

**Reply:** Figure 10 shows an example of the mean SAOD at 525 nm (based on data from 2002-2004) from SAGE II, which is compared with the corresponding SAOD values (520 nm) (no mean values) for January and July. These are shown, for example, in Figures 6 and 8. Since a graphical merge makes no formal sense (also formally incorrect) and the corresponding components are already shown graphically, we decided against this.

**Replies to comments by reviewer 3**

**Comment:** The manuscript addresses one aspect of geoengineering—enhancement of the stratospheric aerosol layer to mitigate climate warming. The authors analyze the detectability of stratospheric aerosols formed as a result of sulphur dioxide ($SO_2$) injection into the stratosphere. The topic is timely and relevant, and the paper could be suitable for publication after some major issues are addressed.

**Reply:** We thank the reviewer for his/her constructive and helpful comments. We tried to answer every comment in an appropriate way.

**Comment:** While the paper is generally well written, further clarification is necessary to ensure the content is accessible to a broader audience beyond specialists in stratospheric aerosols.
The numerical experiment involving $SO_2$ injection should be clearly described in a dedicated section. Specifically:
How was the injection distributed over time? Was it a single injection of 1–2 Tg, or was the amount spread uniformly over the course of a year? The terms "initial phase" and "quasi steady-state phase" should be explicitly defined in this context.
The key distinctions between the geoengineering experiment and isolated volcanic eruptions should be emphasized more clearly.

**Reply:** Thank you for the comment. In the process of addressing the comments from reviewer 1, we added the following illustration and further explanations:
We performed a single simulation over several years. The injections for SAI ran for 15 years. For our study, we took three years at the end of these simulations and averaged them over time. This is similar to previous simulations and publications, e.g. Niemeier et al. (2020) and Weisenstein et al. (2022), where three-year averages were also used. Fig. A1 illustrates the time series of the global sulphate burden showing that the steady-state phase is reached after two years. We used the early phase to include sulphate level below the steady-state level to see if we could detect sulphate even earlier. At this point, the goal was not to use a stabilized result. The aim was to find a lower threshold at which detection would be possible.

[Figure]

**Figure A1.** Monthly mean sulphate burden in kg over time (2005 – 2010) for 1 Tg S/y, showing the differences between the two-year initial phase and the quasi steady-state phase.

As described in the ECHAM section of the preprint, these are continuous injections at an altitude of 60 hPa.

**Comment:** The authors argue that the injection of 1–2 Tg/y of $SO_2$ is relatively small, and therefore, if such an amount is detectable, larger quantities should also be detectable. However, they should clarify what they consider to be the upper limit of $SO_2$ injection. Additionally, they should discuss how detectability changes with increased injection amounts, particularly in light of the potential zero transmittance in occultation geometry, as observed shortly after the Pinatubo eruption.

**Reply:** The upper limit of the injection amount depends on the specific goal. Depending on the model, 8 to 16 Tg SO2 per year would be required to cool the Earth's surface by 1 degree on a global average (Niemeier, 2023).

We assume that with larger injection rates the detectability increases, the aerosol extinction signal becomes larger and the total errors smaller (compare $0\,\text{Tg S/y} \rightarrow 1\,\text{Tg S/y} \rightarrow 2\,\text{Tg S/y}$) up to a certain amount, possibly about 20 Tg (as in the case of the Pinatubo eruption, although a volcanic eruption does not represent continuous injections). We cannot prove it at this point, and it is outside the scope of this study, but we agree that 'zero transmittance' is a potential problem.

However, we note that the zero transmittance problem does not mean that solar occultation measurements are entirely useless. They cannot provide aerosol extinction below a certain altitude, but at slightly higher altitudes they will still work and provide information on enhanced aerosols levels. In the case of Pinatubo, SAGE II measurements were always available at altitudes above about 24 km.

We added this points to the discussions.

Referring to: Niemeier, Ulrike. (2023). Eine künstliche stratosphärische Schwefelschicht: Der einfache Ausweg aus dem Klimaproblem? (Version 1. Aufl.). In WARNSIGNAL-KLIMA: Hilft Technik gegen die Erderwärmung ? Climate Engineering in der Diskussion (pp. 243–249). Hamburg, Germany: Wissenschaftliche Auswertungen in Kooperation mit GEO Magazin-Hamburg. `http://doi.org/10.25592/uhhfdm.12856`

**Comment:** It would also be beneficial to model the particle size distribution and its temporal evolution. This would allow for the calculation of transmittance at wavelengths longer than 550 nm and help overcome the issue of transmittance saturation.

**Reply:** Thank you for the idea. We agree that this is of course a potential problem, but as explained above, it is not relevant for the present study. For the analysis of a 'Pinatubo scenario' we agree it would be a problem, but that is not the case here. We will investigate the effects of S-injections on the particle size distribution of sulphate aerosols - and the effects on detectability - in another study.

**Comment:** Throughout the manuscript, different names are used for the circulation model—ECHAM, EHAM5-HAM, MAECHAM5, and MAECHAM5-HAM. If these refer to the same model, a consistent name should be used. If they are different, the distinctions should be explained.

**Reply:** We adapted this. It is now called MAECHAM-HAM at the first mention (and in the abstract and in the introduction) and thereafter ECHAM.

**Comment:** Table 3: Please justify the use of a 2% value for error estimation.

**Reply:** In the process of addressing the comments of reviewer 1, we added the relevant literature to Table 3. For the temperature and pressure uncertainties: e.g.: Nowlan et al. (2007) and Langland et al. (2008). For the total ozone column: e.g.: Garane et al. (2019) and the pointing error: e.g.: Bramstedt et al. (2012).

Nowlan, C., McElroy, C., and Drummond, J.: Measurements of the O2 A-and B-bands for determining temperature and pressure profiles from ACE–MAESTRO: Forward model and retrieval algorithm, Journal of Quantitative Spectroscopy and Radiative Transfer, 108, 371–388, 2007.

Langland, R. H., Maue, R. N., and Bishop, C. H.: Uncertainty in atmospheric temperature analyses, Tellus A: Dynamic Meteorology and Oceanography, 60, 598–603, 2008.

Garane, K., Koukouli, M.-E., Verhoelst, T., Lerot, C., Heue, K.-P., Fioletov, V., Balis, D., Bais, A., Bazureau, A., Dehn, A., Goutail, F., Granville, J., Griffin, D., Hubert, D., Keppens, A., Lambert, J.-C., Loyola, D., McLinden, C., Pazmino, A., Pommereau, J.-P., Redondas, A., Romahn, F., Valks, P., Van Roozendael, M., Xu, J., Zehner, C., Zerefos, C., and Zimmer, W.: TROPOMI/S5P total ozone column data: global ground-based validation and consistency with other satellite missions, Atmos. Meas. Tech., 12, 5263–5287, https://doi.org/10.5194/amt-12-5263-2019, 2019.

Bramstedt, K., Noël, S., Bovensmann, H., Gottwald, M., and Burrows, J. P.: Precise pointing

knowledge for SCIAMACHY solar occultation measurements, Atmos. Meas. Tech., 5, 2867–2880, https://doi.org/10.5194/amt-5-2867-2012, 2012.

**Comment:** Equations:
Is the sigma ($\sigma$) in Equation 7 the same as r in Equation 6? If so, use consistent notation.

**Reply:** Thank you for the comment. Eq. 7 is a standard expression for determining the total error based on the parameter errors. We understand that the equation can be misinterpreted at first glance and without context, so the text below (line 153) explains: 'The individual errors (relative differences) (compare Eq. 6) of a certain height were added up quadratically.'.

**Comment:** Compare your aerosol extinction error estimates with those from SAGE III aerosol extinction retrievals.

**Reply:** Thank you for this idea. It is difficult to make a comparison, as the present study considers different phases (quasi steady-state phase and initial phase) and different injection rates (1 Tg S/y, 2 Tg S/y and background), which means that only a comparison of the order of magnitude is possible. Wrana et al, 2021 (Tab. 1) shows the extinction measurement uncertainties at 520 nm averaged from June 2017 to December 2019 at an altitude of 20 km (SAGE III/ISS level 2 solar aerosol product) with a value of 5.66 %. The total errors for 1 and 2 Tg S/y at 20 km are of approximately the same order of magnitude for the northern and southern mid-latitudes.

We added this facts to the preprint.

Wrana, F., von Savigny, C., Zalach, J., and Thomason, L. W.: Retrieval of stratospheric aerosol size distribution parameters using satellite solar occultation measurements at three wavelengths, Atmos. Meas. Tech., 14, 2345–2357, https://doi.org/10.5194/amt-14-2345-2021, 2021.

**Comment:** Figures:
A logarithmic scale should be used for figures displaying extinction profiles and optical depths.

**Reply:** We changed that and use now log scales for the extinction profiles (except for Fig. 2). For the figures representing the SAOD, we decided against it, as the clarity of the individual graphs within the figure would be reduced.

**Comment:** Cases with 1 Tg and 2 Tg/year should be shown on the same figure for easier comparison.

**Reply:** Thank you for the suggestion. We tested this, but merging the figures would reduce the clarity of the illustration, reduce the visual selectivity of the individual results and thus impair the interpretation of the central statements at first glance. Nevertheless, we understand the idea and the potential benefits and tested it but decided against it.

**Comment:** In Figure 1, part (a) should represent the initial phase, and part (b) the steady-state

phase, to reflect chronological order. Different colors should be used to indicate high and low extinction values for better visual clarity.

**Reply:** We have changed the order of the subplots (a) and (b) of Figure 1 and adapted the corresponding text. Following the guidelines for illustration in ACP and the specifications for colour blindness friendliness, we have decided to use this colour scale. For the sake of readability of all subplots and due to the different value ranges, it is unfortunately not possible to display all subplots with the same colourbar.

---

## Author Response (AR2)

**Replies to comments by reviewer 2**

**Comment:** Dear Editor, dear Authors,

The manuscript "Investigating the ability of satellite occultation instruments to monitor possible geoengineering experiments" by Lange et al. aims at analysing the expected capability of solar occultation instruments like SAGE III to detect and characterise stratospheric aerosol injection (SAI) geoengineering interventions. As said in first review stage, the topic of the manuscript is potentially important and relevant for a readership of satellite instrument and data scientists. During the first review round, I suggested to review the manuscript around 4 major comments and a number of minor comments. I think that, globally, the Authors satisfactorily clarified these points, even if the changes in the text are quite limited and somewhat incomplete. Thus, I propose to accept the manuscript for publication pending a few further minor changes, which basically are associated with further clarifying these 4 major points + 2 minor points, all detailed in the following.

Regards.

Major Comments:

1) OK got it. I agree that a continuous injection is different from point injection and worth testing in terms of detectability. Unfortunately, this is not yet fully clear from the Introduction/motivation (the proposed change in the Introduction is minimal), so please use more words to explicit this aspect and the underlying motivations of this study.

**Reply:** In the course of responding to the reviewer comments, we have already added the following to the introduction/motivation: "The aim of the current study is to investigate whether it is possible to detect stratospheric aerosols formed from small amounts (in context of possible geoengineering experiments) of sulphur artificially and continuously injected into the stratosphere. Here 1 and 2 Tg S/y, which results in a much lower sulphate injection per time compared to a volcanic eruption with the same injected amount, using a satellite solar occultation instrument, like SAGE III/ISS. "

Nevertheless, we have added the following to the introduction: "Especially in view of the fact that the continuously injected sulphur amounts of 1 and 2 Tg S/y lead to much lower injection amounts per time compared to a volcanic eruption with the same amount injected." (l. 44 – 46)

**Comment:** 2) Despite I don't completely agree with this (AMT sound still more adapted than ACP, for me), I think this is up to the Editor to decide on this.

**Reply:** Thank you for the comment.

**Comment:** 3) Thank you for rewording the sentences I pointed at, this is clearer now. I'm still a bit confused on why "the errors in the background case are larger [than in the initial phase of SAI deployment]": please clarify this in the text

**Reply:** In the course of responding to the the reviewer comments, we explained this relationship below (l. 187 – 188): "...as the signal is stronger at a higher injection rate such as 2 Tg S/y and the total errors are therefore smaller." To avoid possible confusion, we have added the following to the sentence in question, which refers to this sentence (l. 187 – 188): "It should be noted at this point that for the retrievals based on the initial phase data, the errors for the background case are used in the following, based on the assumption that these errors (relative differences) are larger, which is why an error analysis was also carried out for the background case (explanation below).".

**Comment:** 4) The quality of the text and clarity is a bit improved, thanks for following my specific comments

Minor Comments:

1) Minor Comment 7: OK but did you modify the text to clarify this?

**Reply:** We have not adapted the text at this point, as it already says: "The injection of sulphur dioxide into the stratosphere (stratospheric aerosol injection, SAI) is one idea of SRM (e.g., Budyko, 1977; Crutzen, 2006), mimicking the effects of large volcanic eruptions..." (l. 14 – 15).

And:

"SAI is supposed to be relatively cheap (e.g., Moriyama et al., 2017) and companies or start-ups may see an option to earn money with SAI (e.g, "Make sunsets" company (Make sunsets , 2024)). Therefore, it is very important to be able to observe relatively small amounts of sulphur aerosols in the atmosphere." (l. 25 – 28).

**Comment:** 2) Minor Comment 15: OK but has this information been added in the text?

**Reply:** We added this information to the ECHAM chapter: "The solar radiation scheme in ECHAM has 4 spectral bands, 1 for the visible and ultra-violet, and 3 for the near-infrared. The long-wave radiation scheme has 16 spectral bands (see Stier et al. (2005)). Values for single scattering albedo and asymmetry factor are taken from a look-up table, pre calculated with Mie calculations, to save computation time." (l. 76 – 79).

**Replies to comments by reviewer 3**

**Comment:** Figures A2, A3 are not mentioned in the text of the paper.

**Reply:** Thank you for the comment. The figures are mentioned indirectly in the following sentence: "More Figs. for different latitudes can be found in the appendix." (l. 166 – 167). To make the statement clearer, we have added the following to the sentence: "More Figs. for different latitudes can be found in the appendix (Figs. A2, A3)."

**Comment:** Row 183 "he total errors at the altitude" should be "The"

**Reply:** Thank you, changed!